

# Effect of intercropping *Lolium perenne* in *Ziziphus jujuba* orchards on soil quality in the canopy

Yao Ma[1], Bin Cao[1,2], Xiaojia Wang[3] and Weijun Chen[4]

[1] School of Forestry and Pratacultural, Ningxia University, China
[2] Ningxia Technical College of Wine and Desertification Prevention, Yinchuan, Ningxia, China
[3] School of Agriculture, NingXia Polytechnic, Yinchuan, Ningxia, China
[4] Lingwu Natural Resources Bureau, Yinchuan, Ningxia, China

Corresponding author
Bin Cao, bingcao2006@126.com

## ABSTRACT

The cultivation of Lingwu jujube traditionally employs clean tillage, leaving substantial gaps between rows and exposing almost 60% of the orchard to the elements. This method promotes rapid soil moisture evaporation, exacerbates soil erosion, and deteriorates the soil's physicochemical properties. Consequently, there is a critical need for a more sustainable planting approach that optimally utilizes land resources. A pertinent question is whether varying densities of ryegrass intercropping can improve the uptake of nutrients and water by the jujube tree, the primary species in this ecosystem. In this context, a 2-year field experiment was conducted with three densities of perennial ryegrass intercropped with Lingwu jujube. The experiment assessed the impact on soil's physical and chemical attributes beneath the jujube canopy, with a focus on correlating soil moisture, enzyme activity, and physical properties. The findings reveal that intercropping at a medium density most effectively enhanced the soil's physical characteristics. Relative to monoculture, this approach increased the proportion of water-stable aggregates (0.5–0.25 mm) by 4.16%, decreased the soil's fractal dimension by 0.46%, augmented the field water holding capacity by 14.78%, and significantly boosted soil enzyme activity. Furthermore, high-density ryegrass intercropping elevated the soil's organic matter content by 36.09% and ameliorated both the pH and cation exchange capacity. Conversely, low-density intercropping raised soil moisture levels by 40.18% in the top 20 cm of the soil. Collectively, these results suggest that an optimal density of ryegrass in intercropping not only bolsters the moisture retention capabilities of soil in Lingwu jujube orchards but also enhances overall soil fertility. Therefore, the adoption of ryegrass and jujube tree intercropping is highly advisable in the ecologically sensitive and resource-constrained arid sandy regions of northern China, offering substantial practical benefits.

## INTRODUCTION

The challenges to food security have become increasingly urgent and complex due to population growth, urbanization, climate change, and the depletion of agricultural resources (*Gou, 2017*). Agroforestry systems represent a high-yield, efficient, and

sustainable agricultural production model (*Miao, Stewart & Zhang, 2011*). This model improves productivity and profitability by optimizing land use to ensure food security. Agroforestry not only addresses conflicts in land use between agriculture and forestry (*Ma et al., 2022*) and enhances the efficiency of resource utilization both above and below ground (*Zhang et al., 2024*), but also offers numerous additional benefits, such as soil and water conservation (*Ma et al., 2022*), improved soil structure, enhanced soil fertility (*Zhao et al., 2022*), and biodiversity protection (*Ghaffar et al., 2019*).

The primary driving force behind the development of agroforestry systems is the strategic exploitation of the advantages of both trees and crops, minimizing resource competition between species and maximizing resource utilization (*Singh et al., 2011*). Compared to protective forests, economic forests offer more significant economic benefits, which has led to considerable attention being given to their development in the arid and semi-arid regions of northern China over the years. The establishment of economic forests has been prioritized not only for the more effective use of land resources, improvement of soil conditions, and increase in farmers' income, but also for their positive effects on soil and water conservation, thus garnering increasing interest (*Fatima et al., 2024*). As an intensive planting model that optimizes land resources and space to enhance efficiency, agroforestry is crucial in many self-sufficient, low-input, or resource-limited agricultural systems. Numerous studies have demonstrated that intercropping can improve crop yield (*Liu et al., 2024*) and quality (*Liu et al., 2024*) under extreme climatic conditions. However, the establishment of intercropping systems in ecologically fragile regions remains challenging due to constraints related to water resources and economic conditions.

Jujube, a species native to China with over 7,000 years of cultivation history (*Yunmao et al., 2024*), belongs to the Rhamnaceae family. Its fruit contains essential nutrients, such as protein, fat, and vitamin C, and exhibits multiple pharmacological and health benefits (*Li et al., 2024*). The 'Lingwu Changzao' jujube is a high-quality fruit tree variety unique to Ningxia, primarily cultivated along the Yellow River in Lingwu City, Ningxia Hui Autonomous Region. With a cultivation history of 1,300 years, it holds significant economic, ecological, and social value. As the planting area of 'Lingwu Changzao' jujube has expanded annually and the industry has grown, traditional planting methods in the region, which involve clean tillage with extensive bare soil between rows (accounting for nearly 60% of the orchard area), have become increasingly problematic. This method leads to rapid water loss through evaporation, increased soil erosion, and degradation of soil physical and chemical properties. Thus, there is an urgent need for a more effective planting model to optimize land use.

Intercropping forage crops between fruit trees not only makes better use of the orchard land for forage production and suppresses weed growth, but also improves the ecological environment within the orchard. Research has shown that planting perennial plants between fruit trees provides several benefits, including soil and water conservation (*Oliveira et al., 2024*), improved soil fertility (*Silva et al., 2021*; *Zhu et al., 2022*), optimized orchard ecological environments (*Ai, Ma & Hai, 2021*; *Du et al., 2010*), and increased fruit yield (*Rodrigo et al., 2005*) and quality (*Tu et al., 2021*; *Wang et al., 2023*). These practices have been widely adopted in orchard management worldwide.
Soil nutrients, which are essential for plant growth and metabolism, not only influence crop yield but also play a crucial role in the overall soil ecosystem. Recent research on soil quality has included studies on soil physical characteristics, chemical composition, and soil enzyme activity in relation to microbial communities. Soil physicochemical properties are widely recognized as key indicators of soil management effectiveness. However, sustainable soil management in intercropping models—ensuring stable and efficient soil structure that supports root growth (*Shen et al., 2023*) and water retention (*Zheng et al., 2021*)—remains a challenge (*Duan et al., 2024*; *Blaise et al., 2021*). A meta-analysis by *Li et al. (2024)*, involving 2,146 studies on intercropping or mixed cropping and their effects on soil organic matter or carbon, revealed that intercropping increased organic carbon content by 17.75% compared to monocropping. The crop composition and soil depth were identified as major factors influencing soil organic carbon accumulation. Additionally, studies by *Liu et al. (2024)* demonstrated that intercropping legumes improved maize grain yield in nitrogen-deficient environments. Compared to monoculture, maize/legume green manure intercropping significantly increased biocarbon and nitrogen inputs, which, in turn, positively impacted soil enzyme activity and mediated soil carbon, nitrogen, and nutrient cycling, thereby enhancing soil fertility and promoting the growth and nutrient absorption of crops. *Cardinael et al. (2015)* indicated significant variations in soil organic matter content between inter-row and under-tree positions, suggesting that spatial differences between main crops and intercropped species must be fully considered when evaluating the soil physicochemical properties of agroforestry systems. Current research primarily focuses on changes in the physicochemical properties of soil across entire silvopastoral systems, with limited attention to the spatiotemporal variations in soil properties under tree canopies influenced by crops within these systems. Evaluating the impact of silvopastoral systems on soil quality and its spatial differences can help clarify the role of pasture in shaping the orchard's soil microenvironment. However, establishing effective silvopastoral systems in the water-scarce semi-arid and arid regions of northern China and understanding soil moisture consumption within such systems is crucial. Silvopastoral systems can reduce unproductive soil evaporation, promote rainwater retention and infiltration, and potentially improve soil water storage (*Wang et al., 2023*). Conversely, the transpiration of intercrops may increase water consumption in orchards. Intercropping density (*Yin et al., 2020*) is a critical factor in balancing these aspects. For instance, an intercropping experiment by *Song et al. (2020)* with sweet potatoes and walnuts at different densities in southwestern China demonstrated significant differences between densities. Optimizing sweet potato density and spacing improved growth, yield, and photosynthetic traits in the walnut/sweet potato intercropping system. *Xu et al. (2021)* found that, compared to monoculture systems, intercropping significantly reduced maize or alfalfa yield and nitrogen content, though it offered advantages in land and nitrogen use efficiency. Increased nitrogen fertilizer application and maize planting density positively impacted intercropped maize. However, few studies have explored the effects of different intercropping densities on the physicochemical properties of orchard soils. Therefore, evaluating whether different intercropping densities can promote the utilization of

nutrients and soil moisture by the core species of the system—jujube trees—is crucial for assessing the feasibility of intercropping pasture in orchards.

In this study, we established an intercropping system between jujube trees and ryegrass at various densities to determine: (1) whether intercropping affects the physical and chemical properties of the soil beneath the jujube canopy; (2) whether intercropping influences the soil moisture beneath the jujube canopy; and (3) whether changes in soil moisture are associated with soil physical properties. We hypothesized that (1) intercropping enhances the physical and chemical properties of the soil beneath the jujube canopy; (2) intercropping increases the moisture content of the topsoil; and (3) there is a correlation between soil moisture and soil physical properties.

## MATERIALS AND METHODS

### Overview of the study area

The experiment was conducted at the 'Lingwu Changzao' jujube production base of Yinhul Agricultural, Forestry, and Livestock Technology Development Co., Ltd. in Lingwu City, Ningxia (106°23′E, 37°53′N; altitude 1,180 m). The region experiences a typical continental monsoon climate (Fig. 1), with an average annual temperature of 8.9 °C and a cumulative temperature above 0 °C of 3,300 °C. The long-term average annual precipitation is 192.9 mm, with a maximum annual precipitation of 352.4 mm and a minimum of 80.4 mm. The area receives 3,011.0 h of sunshine per year, with an annual evaporation rate of 1,762.9 mm and an average wind speed of 2.6 m/s. The frost-free period ranges from 140 to 160 days. The field soil is sandy in texture, and the vegetation primarily consists of xerophytic and mesoxerophytic plants. Soil physical and chemical indicators before planting area research. The soil capacity was 1.35 g cm$^{-3}$, pH was 8.45, quick-acting nitrogen was 0.41 mg kg$^{-1}$, quick-acting phosphorus was 20.49 mg kg$^{-1}$, quick-acting potassium was 76.84 mg kg$^{-1}$, and organic matter was 4.00 g kg$^{-1}$.

### Experimental materials

The study utilized 'Lingwu Changzao' jujube (*Ziziphus jujuba* Mill. cv. *Lingwu Changzao*), with the jujube trees being 15 years old. The trees were planted in a north-south orientation with a spacing of 2 × 6 m. During the 2023–2024 period, the trunk diameter of the 'Lingwu Changzao' jujube trees ranged from 105 to 145 mm, the canopy height ranged from 3.1 to 3.8 m, and the canopy width ranged from 3.2 to 4.1 m. The selected perennial ryegrass varieties and their sources are listed in Table 1.

### Experimental design and field management

Building on previous experiments, perennial ryegrass was selected from nine forage species, including perennial ryegrass, matgrass, tufted hairgrass, flat-stemmed meadow-grass, purple-flowered alfalfa, red clover, long-soft-haired wild pea, and white clover, for further study (*Xiaojia, 2023*). Perennial ryegrass was chosen based on its strong root system, good growth under fruit trees, faster germination and coverage compared to other species, preference for warmth, shade tolerance, and strong resilience (*Wang et al., 2023*). The experiment followed a completely randomized block design. Two systems were

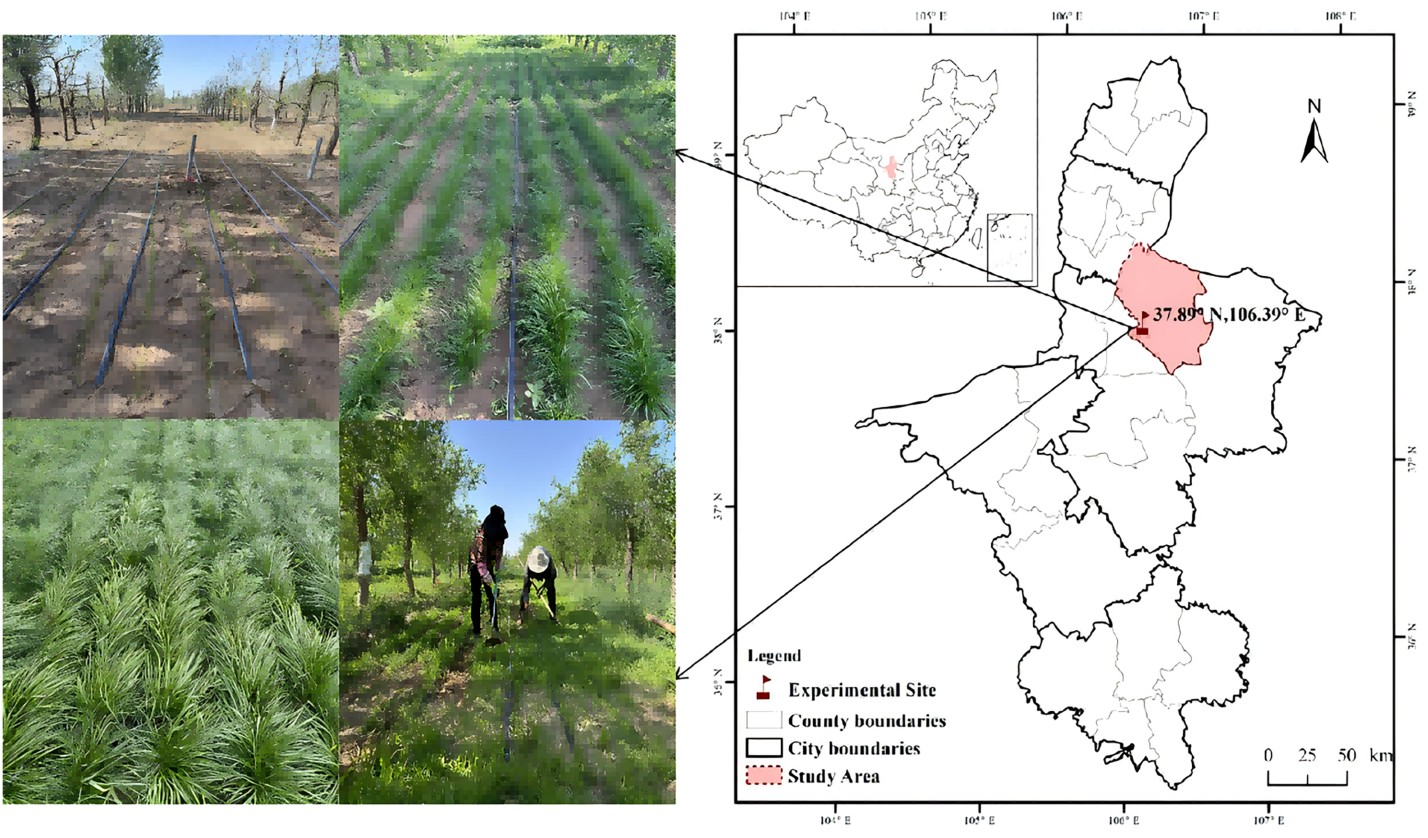

**Figure 1 Test site.**

**Table 1 Sources of test varieties.**

| Material | Cultivar | Seed purity | Ratio of germination | Source |
|---|---|---|---|---|
| Perennial ryegrass (*Lolium perenne* L.) | Newyork | 98% | 85% | Ningxia Fenglv Agroforestry Co. |

established in the jujube orchard: 'Lingwu Changzao' jujube intercropped with perennial ryegrass (J–R) and 'Lingwu Changzao' jujube monoculture (CK). Previous research identified the optimal ryegrass planting density as 15–30 kg hm$^{-2}$ (*Mengyan et al., 2023*). Based on this, three treatments were selected: M1 = 15 kg hm$^{-2}$ (low), M2 = 22.5 kg hm$^{-2}$ (medium), and M3 = 30 kg hm$^{-2}$ (high). Each treatment was replicated three times, resulting in a total of 12 plots. Each plot measured 3 m × 5 m × 2 = 30 m$^2$, with a 5 m gap between plots to serve as a natural grass isolation area. Each replicate in the intercropping system included three rows of jujube trees, with three trees per row. The jujube tree bases were covered with black plastic mulch, with a width of 1 m, and the mulch was placed 1 m away from the edge of the grass strip. The inspection area design is as shown in Fig. 2.

## Measurement indicators and methods

Before the 2023 grass sowing season, the soil was tilled using a rotary tiller to depths between 20 and 30 cm to eliminate weeds, followed by sowing in rows 30 cm apart and at a
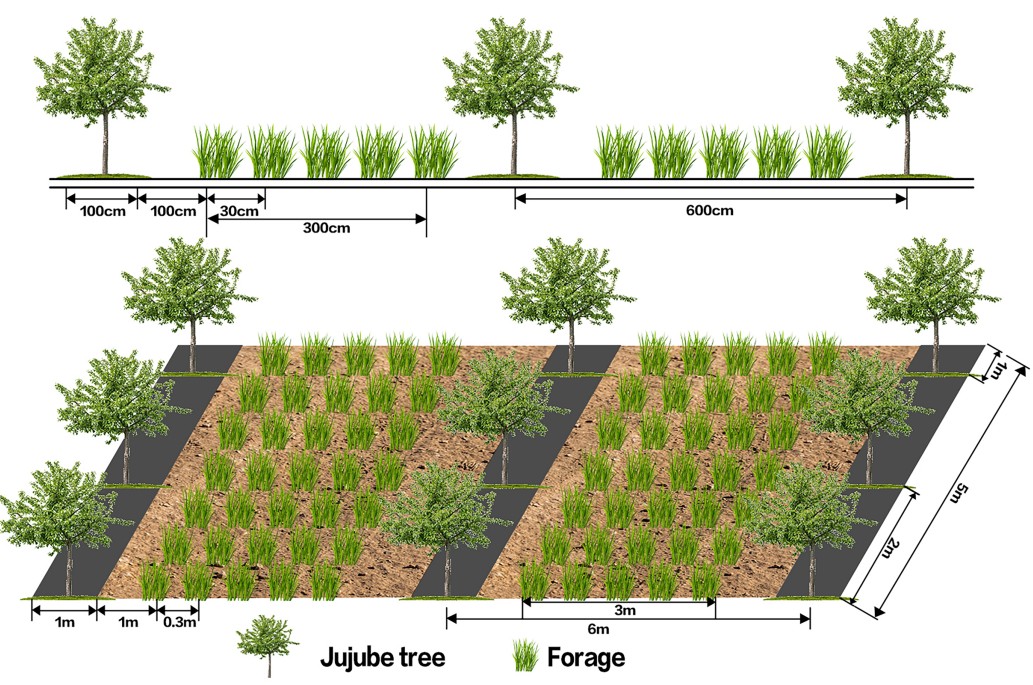

**Figure 2 Schematic diagram of experimental design.**

depth of about 2–3 cm. In the monoculture system of Lingwu jujube, weeding was carried out twice monthly through both mechanical and manual methods, and debris was consistently cleared to preserve a clean-tilled condition. Grass was harvested twice per year, maintaining a stubble height of 5 cm. The first harvest occurred between June 25 and 27, and the second between August 8 and 10, coinciding with the ryegrass earing stage. Throughout the experiment, management practices across the orchard were uniform except in the test zones. Lingwu jujube trees were fertilized with humic acid bio-organic fertilizer (3,000 kg ha$^{-1}$) and water-soluble fertilizers containing nitrogen, phosphorus, and potassium (N 57%, P2O5 46%, K2O 50%; 200–230 kg ha$^{-1}$), supplemented with trace element fertilizers (6–10 kg ha$^{-1}$). Fertilizers were applied by digging a circular trench 30 cm deep and 20 cm wide around the center of each tree, subsequently covered with soil. No fertilization was applied between the rows of jujube trees or in the grass-only systems. During the plant growth stages, a micro-sprinkler irrigation system was implemented in the spaces between rows, with an annual irrigation allocation of 2,750 to 3,250 m$^3$ ha$^{-1}$.

## Soil physical and chemical properties

At the end of August each year, soil physical properties were measured in four soil layers: 0–5 cm, 5–10 cm, 10–20 cm, and 20–40 cm. Sampling points were set at a distance of 1 m on either side of the jujube tree roots under the tree canopy in each plot. A total of 2 sampling points were selected beneath each tree. Samples were collected using the ring knife method and then brought to the laboratory for analysis. Each sample was divided into two parts. The first part was air-dried and used to determine soil physical and

chemical properties. The second part was stored in a refrigerator at 4 °C for the determination of soil moisture, available nutrient content, nitrogen content, and soil enzyme activity.

The primary soil physical properties measured were soil bulk density, saturated hydraulic conductivity, and field water holding capacity. Soil bulk density was determined using the ring knife method, where the outdoor samples were dried to a constant weight using the oven-drying method, and the bulk density of each soil layer was calculated using a specific formula. Saturated hydraulic conductivity was measured using the constant head method (5 cm), and field water holding capacity was measured using an indoor method.

Total soil porosity (TP) was calculated based on the relationship between soil bulk density and particle density, with the particle density (PD) assumed to be the commonly used value of 2.65 g/cm$^3$. Capillary porosity (CP) was calculated based on the relationship between field water holding capacity and soil bulk density. Non-capillary porosity (NCP) was determined from the relationship between total porosity and capillary porosity. The specific formulas are as follows:

$$TP = \left(1 - \frac{BD}{PD}\right) \times 100\% \tag{1}$$

$$CP = FC \times BD \times 100\% \tag{2}$$

$$NCP = TP - CP. \tag{3}$$

TP represents total soil porosity, BD represents bulk density, PD represents particle density, CP represents capillary porosity, FC represents field capacity, and NCP represents non-capillary porosity.

Soil moisture content (SWC) was determined by drying the samples in an oven at 105 °C for 48 h. Bulk density (BD) was measured using the ring knife method (*Shengguo, 2019*). The content of water-stable aggregates in the soil was determined by first dry-sieving 500 g of air-dried soil. The mass proportions of the soil aggregates obtained from dry sieving were used to prepare a 50 g soil sample, which was then subjected to the Savinov wet sieving method to measure the content of water-stable aggregates in different size classes: >5 mm, 2–5 mm, 1–2 mm, 0.5–1 mm, 0.25–0.5 mm, and <0.25 mm. The stability of the aggregates was assessed using fractal dimension (D), mean weight diameter (MWD), and geometric mean diameter (GMD). The calculation methods for these indices were based on the model established by *Lee et al. (2011)*.

Soil pH was measured using a glass electrode in soil-water suspensions with ratios of 1:2.5 and 1:5. Total nitrogen (TN) was determined by the semi-micro Kjeldahl method. Total phosphorus (TP) and total potassium (TK) were measured using the molybdenum-antimony colorimetric method and flame photometry, respectively. Available phosphorus (AP) was determined by extracting the soil with 0.5 mol $NaHCO_3$ followed by molybdenum-antimony colorimetry. Available potassium (AK) was measured by extracting the soil with $NH_4OAC$ and then using flame photometry.

**Table 2 Soil enzymes and their corresponding functions, substrates, and enzyme commission number.**

| Enzyme | Enzyme function in soils | Substrate | EC[a] |
|---|---|---|---|
| β-glucosidase (BG) | Hydrolysis of cellulose | 4-MUB-β-D-glucopyranoside | 3.2.1.21 |
| Cellobiohydrolase (CBH) | Hydrolysis of cellulose | 4-MUB-β-D-cellobioside | 3.2.1.91 |
| Alkaline phosphatase (AKP) | Hydrolysis of phospho-monoesters | 4-MUB-phosphate | 3.1.3.1 |
| Urease (UR) | Hydrolysis of carbamide | Urea | 3.5.1.5 |

Cation exchange capacity (CEC) and soil organic carbon (SOC) (*Snap et al., 2008*) were measured using the external heating method with potassium dichromate. SOC was determined after dissolving $CaCO_3$ with 2 mol $dm^{-3}$ HCl and decomposing organic matter with 30% $H_2O_2$. Soil particle size distribution (sand, silt, and clay) was determined by dispersing the samples with $Na(PO_3)_6$, followed by measurement using the pipette method. $NH_4^+$ in the soil was quantified by steam distillation, and the cation exchange capacity (CEC) was subsequently determined.

Additionally, for each soil layer (0–5 cm, 5–10 cm, 10–20 cm, and 20–40 cm), the soil samples were thoroughly mixed, and the SOC, CEC, and pH were measured. These measurements were conducted in the laboratory, where SOC was assessed using the external heating method with potassium dichromate. After dissolving $CaCO_3$ with 2 mol $dm^{-3}$ HCl and decomposing organic matter with 30% $H_2O_2$, the soil particle size distribution (sand, silt, and clay) was determined using the pipette method. $NH_4^+$ was quantified by steam distillation to determine the soil's cation exchange capacity (CEC).

## Soil enzyme activity

In this study, we measured the activities of three enzymes involved in the carbon, nitrogen, and phosphorus cycles using microplate and fluorogenic substrate methods. Urease (UR) activity was determined using the indophenol blue colorimetric method. Enzyme activities are expressed in nmol $g^{-1}$ $h^{-1}$. The functions of the enzymes and their corresponding fluorogenic substrates are presented in Table 2.

## Nutrient effects in the intercropping system

The introduction of ryegrass into the orchard resulted in spatial overlap of soil moisture and nutrients. On one hand, the soil nutrients became a shared resource between the jujube trees and ryegrass, leading to competition and a reduction in soil nutrient content. On the other hand, ryegrass contributed to improving the soil beneath the jujube tree canopy, providing a beneficial effect on soil nutrients. The combined effects of nutrient reduction and enhancement created a new balance in soil nutrient content. The degree to which the crop impacts the soil nutrients in the jujube orchard can be quantified as the nutrient effect (%) of the agroforestry intercropping system, which can be calculated using Eq. (4).

$$E_N = (SN - SN_{CK})/SN_{CK} \times 100\%. \tag{4}$$

SN represents the average soil nutrient mass fraction in a specific soil layer during the crop growth period in the intercropping system. $SN_{cx}$ represents the corresponding soil nutrient mass fraction in the same soil layer of the control monocropping system. $E_N$ represents the nutrient effect of the soil.

## Data analysis

Data analysis was performed using SPSS 26.0 (SPSS Inc., Armonk, NY, USA). One-way analysis of variance (ANOVA) was used to compare differences in soil water-stable aggregates, soil nutrient content, and soil enzyme activity under different planting patterns. Duncan's multiple range test was employed to assess the significance of differences between treatment means at the $p = 0.05$ level. Two-way ANOVA was used to examine the combined effects of different planting patterns, soil layers, and their interactions on various parameters. All graphs and charts were created using Origin 2024 (Version 10.1, Origin Lab Corporation, Northampton, MA, USA). Data in the tables are presented as mean ± standard error.

# RESULTS

## Results and analysis

### *Changes in the characteristics of water-stable aggregates in the soil beneath the jujube tree canopy*

Figure 3 shows the distribution characteristics of water-stable aggregates in the soil beneath the jujube tree canopy under varying densities of ryegrass intercropping systems. During the 2-year experiment, the dominant size class of water-stable aggregates in all soil layers across various ryegrass densities and 'Lingwu Changzao' jujube intercropping systems was the <0.25 mm microaggregate class, with contents ranging from 79.70% to 91.08%. The 0.5–0.25 mm and 1–0.5 mm aggregate classes followed, with contents of 6.65–12.87% and 0.78–1.59%, respectively. Aggregates larger than 5 mm had the lowest content, ranging from 0% to 3.58%. "In the 0–5 cm soil layer, the content of <0.25 mm water-stable aggregates was highest in the high-density ryegrass and 'Lingwu Changzao' jujube intercropping system during both 2023 and 2024, showing an increase of 2.86% to 11.36%. The 0.5–0.25 mm water-stable aggregate content was highest in the medium-density ryegrass and jujube intercropping system, increasing by 2.12% to 6.25%. Similarly, the 1–0.5 mm water-stable aggregate content was highest in the medium-density system. "In the 5–10 cm soil layer, the content of <0.25 mm water-stable aggregates was highest in both the high-density ryegrass-jujube intercropping system and the monoculture jujube system, with contents of 88.24% and 88.31%, respectively. The content of 0.5–0.25 mm water-stable aggregates was highest in the medium-density system, as was the content of 1–0.5 mm aggregates. With increasing soil depth, the proportion of <0.25 mm water-stable aggregates gradually increased across the low-density, medium-density, and monoculture systems during 2023 and 2024. However, the proportion of 0.5–0.25 mm aggregates decreased in both the low-density and medium-density systems.

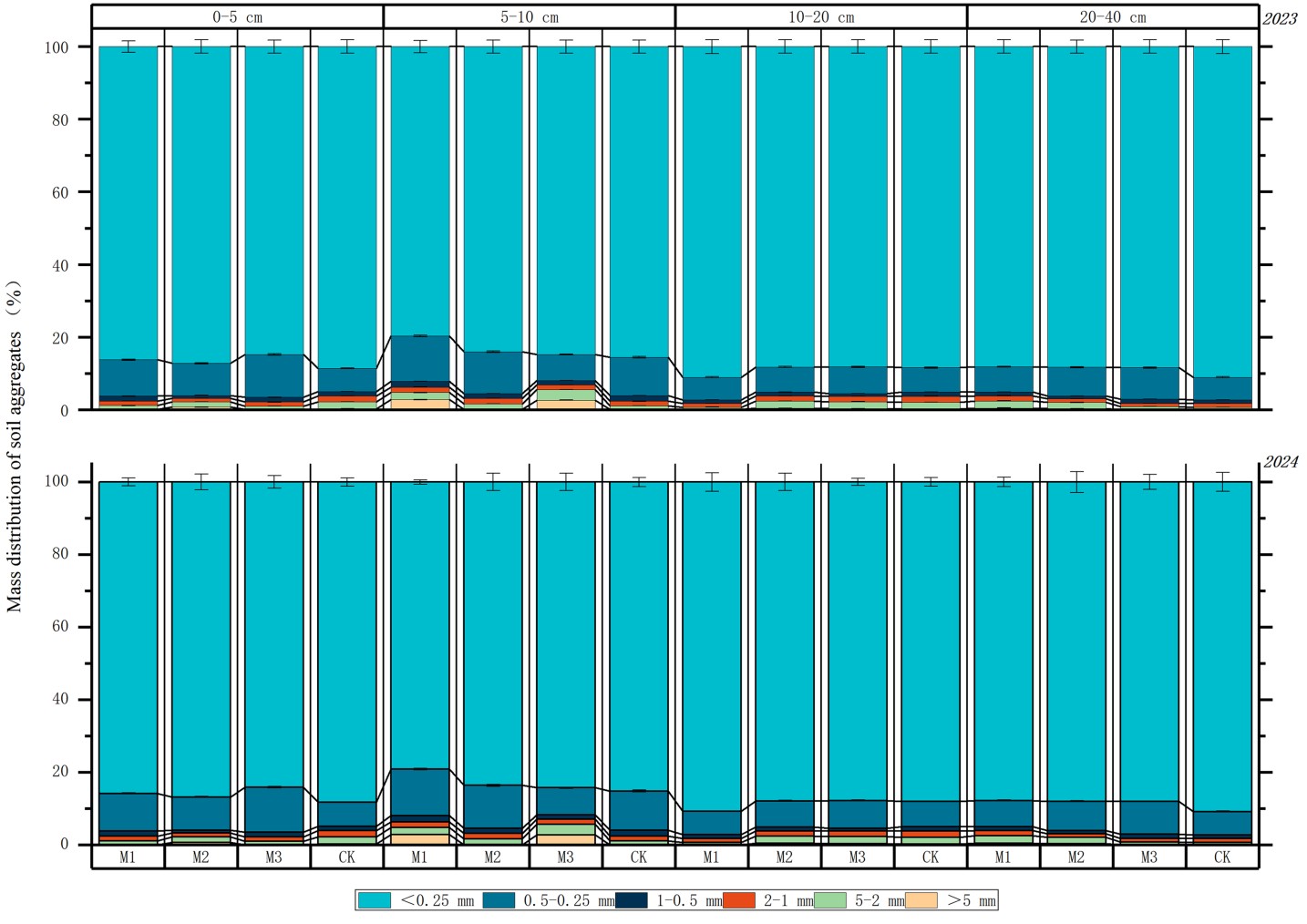

**Figure 3 Distribution characteristics of soil water stability clusters in different cropping systems in 2023–2024.** Note: M1: Lingwu jujube-low-density perennial ryegrass intercropping; M2: Lingwu jujube-medium-density perennial ryegrass intercropping; M3: Lingwu jujube-high-density perennial ryegrass intercropping; CK indicates Lingwu jujube monocropping (Sample size $n$ = 3).

The proportion of 1–0.5 mm aggregates showed no significant variation across the three intercropping systems or the monoculture system. The proportions of 1–0.5 mm and 2–1 mm aggregates in different soil layers across various intercropping systems varied slightly, ranging from 0.45% to 4.31%. The combined proportion of these two size classes remained around 4.50%."

Figure 4 shows the fractal dimension (D), mean weight diameter (MWD), and geometric mean diameter (GMD) of the soil in various ryegrass density and 'Lingwu Changzao' jujube intercropping systems. The fractal dimension of the soil in the intercropping systems was reduced by 0.18–0.19%, 0.45–0.47%, and 0.02–0.04%, respectively, compared to the monoculture system. Among the treatments, the medium-density system had the lowest fractal dimension across all soil layers. The results for 2023 and 2024 were relatively consistent within the treatments. However, the D values for all planting densities were higher in 2024 than in 2023, increasing by 0–0.34%. For

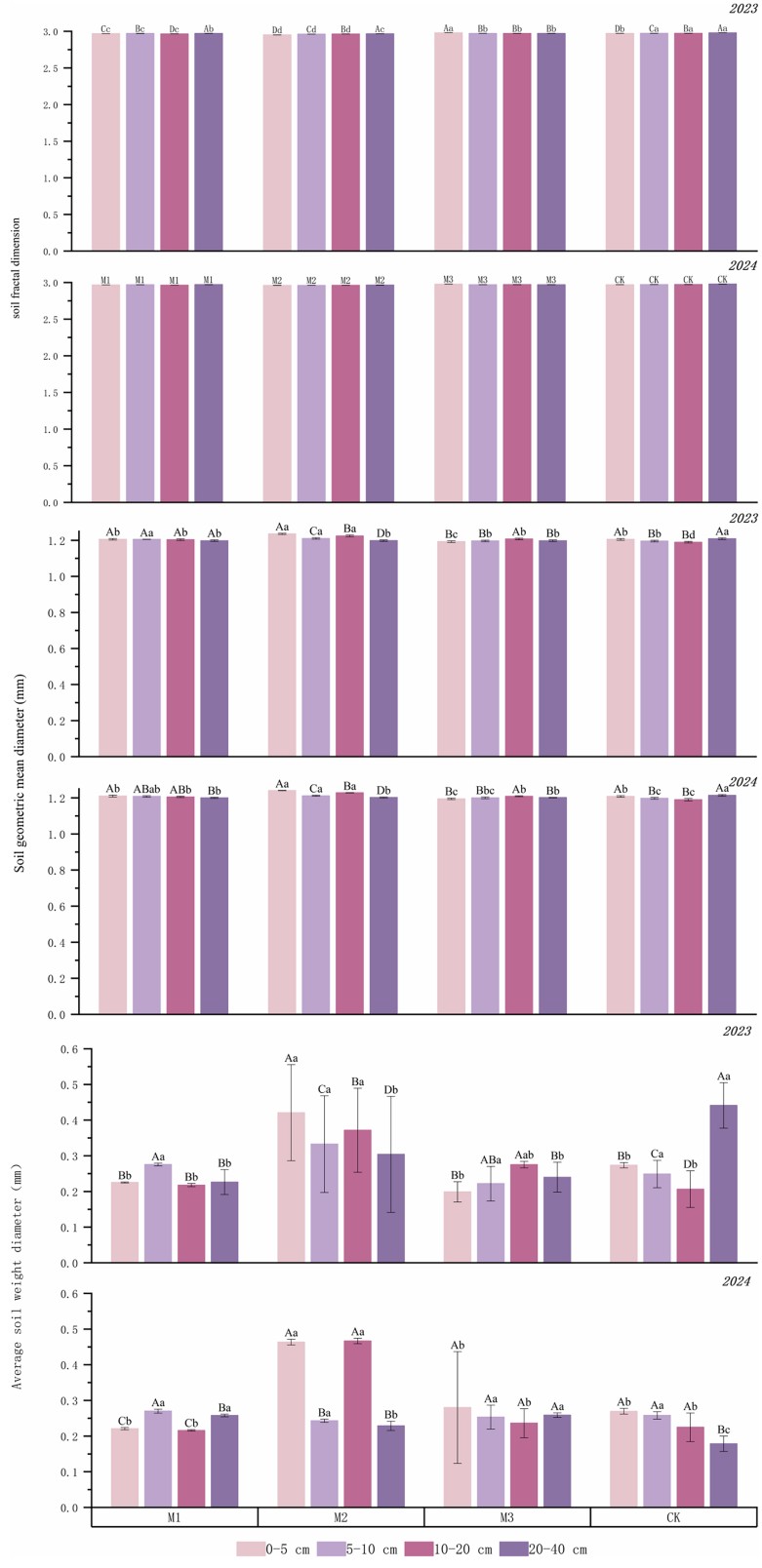

**Figure 4 Evaluation parameters of soil water stability aggregates under different cropping patterns in 2023-2024.** M1: Lingwu jujube-low-density perennial ryegrass intercropping; M2: Lingwu jujube-medium-density perennial ryegrass intercropping; M3: Lingwu jujube-high-density perennial ryegrass

**Figure 4** (continued)
intercropping; CK indicates Lingwu long jujube monoculture; different capital letters indicate significant differences ($p < 0.05$) in different soil layers at the same density; different lowercase letters indicate significant differences ($p < 0.05$) in the same soil layer at different densities. (Values are means of three repetitions ± standard error).               

MWD, the various ryegrass density and jujube intercropping systems showed increases of 3.62%, 50.63%, and 10.47% compared to the monoculture system in 2023. However, in 2024, only the medium-density system showed an increase of 22.11% compared to the monoculture system, while the low- and high-density systems decreased by 19.31% and 20.04%, respectively. The 2-year data indicated that MWD changes within different soil layers were relatively consistent within the same treatment. However, the MWD in the 20–40 cm soil layer of the monoculture system in 2024 was 21% lower than in 2023. For GMD, the three ryegrass density and jujube intercropping systems showed increases of 0.13–0.23%, 1.39–1.50%, and 1.42–1.50%, respectively, compared to the monoculture system. The 2-year data consistently showed that the medium-density system had the highest GMD increase compared to the monoculture system. The GMD in the monoculture system showed a decreasing trend across different soil layers, whereas changes in GMD among the different soil layers in the low-density system were not significant ($p = 0.969$).

### Changes in soil bulk density, field water holding capacity, and saturated hydraulic conductivity beneath the jujube tree canopy

As shown in Table 3, the soil bulk density, saturated hydraulic conductivity, total porosity, and non-capillary porosity beneath the jujube tree canopy varied significantly. among different densities of ryegrass intercropping systems and the control during the 2-year experiment. Compared to monoculture, all three intercropping systems increased field water holding capacity and saturated hydraulic conductivity to varying extents. Although there was a slight increasing trend in soil bulk density over the 2-year period, the differences were minimal. The average soil bulk density ranged from 1.36 to 1.40 g cm$^{-3}$ in the low-density ryegrass intercropping system, 1.31 to 1.40 g cm$^{-3}$ in the medium-density system, and 1.37 to 1.39 g cm$^{-3}$ in the high-density system. In comparison, the average soil bulk density in the monoculture system ranged from 1.38 to 1.42 g cm$^{-3}$. No significant differences were observed among the three intercropping treatments. Field water holding capacity is a key indicator of soil water retention performance. The study found that all three intercropping systems significantly increased the soil's field water holding capacity. In the low-density ryegrass intercropping system, the average field water holding capacity under the jujube trees was 14.60 and 14.79 cm$^3$ cm$^{-3}$ over the 2 years, representing increases of 12.30% and 6.80%, respectively, compared to the monoculture system. In the medium-density system, the average field water holding capacity was 15.08 and 14.48 cm$^3$ cm$^{-3}$, with increases of 16.00% and 4.60%. In the high-density system, the average field water holding capacity was 14.60 and 14.88 cm$^3$ cm$^{-3}$, with increases of 12.3% and 7.5% compared to the monoculture system. Overall, field water holding capacity did not significantly change with increasing soil depth, and in all three intercropping systems,

**Table 3 Changes in soil physical properties under the canopy of date palm under different treatments.**

| Treatments | Soil layers (cm) | Bulk density (g cm$^{-3}$) | | Field capacity (%) | | Hydraulic conductivity (cm d$^{-1}$) | |
|---|---|---|---|---|---|---|---|
| | | 2023 | 2024 | 2023 | 2024 | 2023 | 2024 |
| M1 | 0–5 | 1.33 ± 0.02Aab | 1.43 ± 0.02Aa | 14.03 ± 0.04Bab | 15.57 ± 0.41Aa | 58.75 ± 1.32Ab | 49.36 ± 0.95Aa |
| | 5–10 | 1.36 ± 0.03Aa | 1.43 ± 0.03Aa | 14.85 ± 0.35Aa | 14.45 ± 0.32Ba | 49.63 ± 2.51Bb | 47.23 ± 0.27Ba |
| | 10–20 | 1.36 ± 0.08Aa | 1.37 ± 0.03Bb | 14.45 ± 0.55ABa | 14.50 ± 0.26Ba | 42.30 ± 2.45Cb | 46.31 ± 0.74Ba |
| | 20–40 | 1.38 ± 0.05Ab | 1.36 ± 0.03Ba | 15.05 ± 0.25Aa | 14.63 ± 0.14Bb | 36.10 ± 2.52Dc | 36.59 ± 0.55Ca |
| M2 | 0–5 | 1.27 ± 0.05Ab | 1.33 ± 0.02Aa | 15.07 ± 0.82Aa | 15.32 ± 0.13Aa | 69.57 ± 4.92Ba | 47.05 ± 0.65Ab |
| | 5–10 | 1.31 ± 0.07Aa | 1.43 ± 0.02Aa | 15.81 ± 1.00Aa | 14.25 ± 0.31BCa | 65.32 ± 4.28Ba | 39.09 ± 0.46Bd |
| | 10–20 | 1.32 ± 0.07Aa | 1.35 ± 0.01Cb | 14.91 ± 0.35Aa | 14.40 ± 0.23Ba | 53.01 ± 3.71Aa | 40.18 ± 0.23Bb |
| | 20–40 | 1.33 ± 0.06Ab | 1.37 ± 0.02Ba | 14.56 ± 0.61Ab | 13.95 ± 0.05Ca | 47.21 ± 2.31Ab | 34.14 ± 0.82Cb |
| M3 | 0–5 | 1.38 ± 0.05Aa | 1.43 ± 0.02Aa | 14.98 ± 0.63Aa | 14.42 ± 0.02Cb | 44.09 ± 2.79Bc | 46.58 ± 0.94Ab |
| | 5–10 | 1.39 ± 0.07Aa | 1.42 ± 0.03Aa | 14.65 ± 0.42Aa | 14.57 ± 0.26BCa | 40.57 ± 3.24Bc | 42.47 ± 1.27Bc |
| | 10–20 | 1.41 ± 0.04Aa | 1.34 ± 0.03Bb | 14.97 ± 0.30Aa | 14.85 ± 0.07Ba | 34.70 ± 2.42Cc | 39.23 ± 0.51Cb |
| | 20–40 | 1.36 ± 0.08Aab | 1.38 ± 0.033Ba | 13.86 ± 1.29Ac | 15.68 ± 0.27Ac | 52.12 ± 3.65Aa | 33.59 ± 0.84Db |
| CK | 0–5 | 1.42 ± 0.06Aa | 1.34 ± 0.04ABb | 13.15 ± 0.77Ab | 13.40 ± 0.31Bc | 41.98 ± 2.54Ac | 49.31 ± 0.89Aa |
| | 5–10 | 1.39 ± 0.05Aa | 1.36 ± 0.04Bb | 12.91 ± 0.70Ab | 13.56 ± 0.43Bb | 39.85 ± 2.79ABc | 45.12 ± 0.47Bb |
| | 10–20 | 1.43 ± 0.08Aa | 1.42 ± 0.02Aa | 12.32 ± 0.90Ab | 13.78 ± 0.36Bb | 38.54 ± 2.54ABbc | 45.06 ± 1.13Ba |
| | 20–40 | 1.42 ± 0.02Aa | 1.39 ± 0.03Ba | 13.44 ± 0.38Ac | 14.62 ± 0.27Ab | 36.74 ± 0.47Bc | 35.23 ± 0.85Cab |
| Treatments | | ** | * | *** | *** | *** | *** |
| Soil layers | | ns | *** | ns | *** | *** | *** |
| Treatments × Soil layers | | ns | *** | ** | *** | *** | *** |

**Note:**
M1: Lingwu jujube-low-density perennial ryegrass intercropping; M2: Lingwu jujube-medium-density perennial ryegrass intercropping; M3: Lingwu jujube-high-density perennial ryegrass intercropping; CK indicates Lingwu jujube monocropping; different upper-case letters in the same column indicate that the difference between different soil layers under the same density is significant ($p < 0.05$); different lower-case letters in the same column indicate that the difference between the same soil layers under the different density is significant ($p < 0.05$). *, **, *** and ns represent two-way ANOVA $p < 0.05$, $p < 0.01$, $p < 0.001$ and not significant, respectively. (Values are means of three repetitions ± standard error).

it was significantly higher than in the monoculture system at the same soil depth ($p < 0.01$). Saturated hydraulic conductivity, crucial for determining the dynamic movement of water through saturated soil, showed notable differences across the systems. In the low-density ryegrass intercropping system, the average saturated hydraulic conductivity under the jujube tree canopy was 46.7 and 44.87 cm d$^{-1}$ over the 2 years. In the medium-density system, the values were 58.78 and 40.12 cm d$^{-1}$, while in the high-density system, they were 42.87 and 40.47 cm d$^{-1}$. All three intercropping systems significantly increased average saturated hydraulic conductivity compared to the monoculture system (39.28 and 43.68 cm d$^{-1}$). The medium-density ryegrass intercropping system demonstrated the best performance.

Soil porosity is a critical indicator of soil physical properties, directly influencing permeability, aeration, and water retention capacity, which in turn affect plant growth and soil fertility. During the 2-year experiment, soil porosity varied significantly (Fig. 5). In 2023, no significant differences in soil porosity were observed beneath the jujube tree canopy across the different treatments. Total soil porosity was 48.37% in the low-density

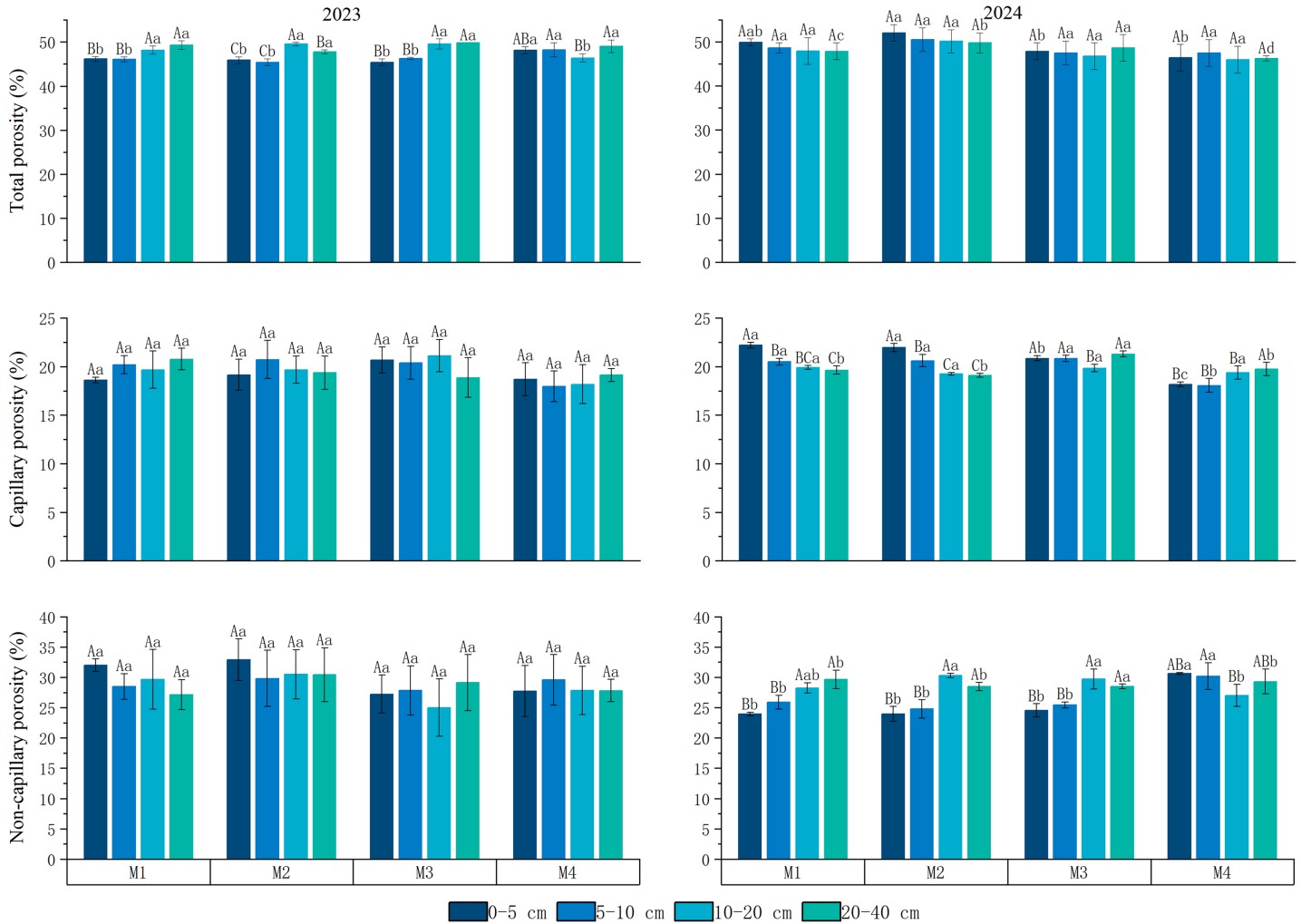

**Figure 5 Changes in soil physical properties under the canopy of date palm under different treatments.** M1: Lingwu jujube-low-density perennial ryegrass intercropping; M2: Lingwu jujube-medium-density perennial ryegrass intercropping; M3: Lingwu jujube-high-density perennial ryegrass intercropping; CK indicates Lingwu jujube monocropping; different upper-case letters in the same column indicate that the difference between different soil layers under the same density is significant ($p < 0.05$); different lower-case letters in the same column indicate that the difference between the same soil layers under the different density is significant ($p < 0.05$). (Values are means of three repetitions ± standard error).

ryegrass and 'Lingwu Changzao' jujube intercropping system, 47.48% in the medium-density system, 48.05% in the high-density system, and 48.34% in the monoculture system. No notable differences were found across different soil layers in any of the treatments. However, in the 2024 experiment, total soil porosity in the low-, medium-, and high-density ryegrass intercropping systems was 2.32%, 2.62%, and 1.06% higher, respectively, than in the monoculture system (48.03%). For capillary porosity, the 2-year average in the low-density ryegrass intercropping system was 19.99%, which was 1.37% higher than in the monoculture system. In the medium-density system, the average was 20.29%, 1.67% higher than in the monoculture system, while in the high-density system, the average was 20.20%, 1.58% higher than in the monoculture system. In the 2023 experiment, capillary porosity showed no significant differences ($p = 0.057$) across

different soil layers within the treatments. However, in 2024, the low-density, medium-density, and high-density ryegrass intercropping systems showed a significant decreasing trend in capillary porosity with increasing soil depth (0–5 cm > 5–10 cm > 10–20 cm > 20–40 cm). For non-capillary porosity, no significant differences were found between the various ryegrass densities in the intercropping systems and the monoculture system.

### Changes in total and available nitrogen, phosphorus, and potassium in soil beneath the jujube tree canopy

The total contents of nitrogen (N), phosphorus (P), and potassium (K) in the soil, along with the levels of available nutrients, are directly related to the growth of jujube trees. Therefore, we analyzed changes in soil nutrient content beneath the jujube tree canopy to explore the nutrient dynamics in the intercropping systems (Table 4). Different intercropping densities and soil layers significantly affected total nitrogen, total phosphorus, total potassium, alkaline-hydrolyzable nitrogen, available phosphorus, and available potassium ($p = 0.031$). The interaction between intercropping density and soil layers also significantly influenced these parameters ($p = 0.017$). Overall, comparing the three intercropping densities to the monoculture system showed no significant increase or decrease in total nitrogen content. However, in the shallow soil layer (0–10 cm), all three intercropping systems had higher total nitrogen content than the monoculture system. Additionally, in the 10–40 cm soil layer, total nitrogen content was higher in the medium- and high-density ryegrass intercropping systems than in the monoculture system. For total phosphorus content, the average in the 0–40 cm soil layer was 2.40% to 18.75%, 28.13% to 44.70%, and 16.87% to 38.54% higher in the monoculture system compared to the low-, medium-, and high-density ryegrass intercropping systems, respectively. The effect on alkaline-hydrolyzable nitrogen varied between years. In 2023, the average alkaline-hydrolyzable nitrogen content in the 0–40 cm soil layer was 49.49%, 45.73%, and 57.06% higher in the low-, medium-, and high-density ryegrass intercropping systems, respectively, compared to the monoculture system. However, in 2024, the average alkaline-hydrolyzable nitrogen content in the monoculture system was 25.88%, 19.24%, and 26.13% higher than in the low-, medium-, and high-density ryegrass intercropping systems, respectively. Overall, the medium- and high-density ryegrass intercropping systems had significantly lower available phosphorus content compared to the monoculture system, with average decreases of 12.04% and 27.73% over the 2-year experiment. Available potassium content in the monoculture system was significantly higher than in the low- and medium-density ryegrass intercropping systems. However, the high-density ryegrass intercropping system had an available potassium content 5.49% higher than in the monoculture system.

Soil organic matter content is widely recognized as a key indicator of soil quality. It enhances soil nutrient supply and retention capacity, thereby improving nutrient availability. Moreover, it promotes the formation of soil aggregates, improving permeability, aeration, water retention, and soil buffering capacity. As shown in Table 5, intercropping significantly increased soil organic matter content compared to monoculture ($p < 0.05$). Overall, the average organic matter content in the 0–40 cm soil

**Table 4 Soil total and quick nutrient content under the canopy of date palm in different treatments.**

| Treatments | Soil layers (cm) | Total N (g/kg) | | Total P (g/kg) | | Total K (g/kg) | | Alkali hydrolyzed nitrogen (mg/kg) | | Avail. P (mg/kg) | | Avail. K (mg/kg) | |
|---|---|---|---|---|---|---|---|---|---|---|---|---|---|
| | | 2023 | 2024 | 2023 | 2024 | 2023 | 2024 | 2023 | 2024 | 2023 | 2024 | 2023 | 2024 |
| M1 | 0–5 | 0.44 ± 0.02Ab | 0.50 ± 0.01Bc | 0.72 ± 0.02Aa | 0.53 ± 0.01Cc | 17.96 ± 0.20Ab | 17.29 ± 0.33Ab | 57.7 ± 1.10Aa | 11.41 ± 0.13Ab | 7.00 ± 0.19Ac | 7.08 ± 0.13Dc | 47.25 ± 1.38Aa | 53.95 ± 1.89Ac |
| | 5–10 | 0.42 ± 0.02Ab | 0.62 ± 0.02Aa | 0.68 ± 0.03Aa | 0.62 ± 0.02Ba | 17.71 ± 0.22ABb | 16.77 ± 0.29ABb | 43.26 ± 1.04Ba | 8.12 ± 0.17Bd | 6.40 ± 0.09Bc | 10.43 ± 0.21Cb | 41.66 ± 0.81Bd | 50.75 ± 1.77ABc |
| | 10–20 | 0.31 ± 0.03Bc | 0.41 ± 0.02Cb | 0.72 ± 0.03Aa | 0.51 ± 0.01Cb | 17.76 ± 0.16ABb | 16.22 ± 0.45Ba | 16.83 ± 0.59Cc | 7.59 ± 0.13Cd | 5.80 ± 0.12Cb | 11.84 ± 0.07Ba | 41.03 ± 0.40Bd | 53.29 ± 1.87Ab |
| | 20–40 | 0.22 ± 0.03Cc | 0.47 ± 0.02Ba | 0.58 ± 0.02Bb | 0.68 ± 0.02Ab | 17.54 ± 0.07Ba | 16.20 ± 0.18Ba | 12.23 ± 0.56Dc | 6.49 ± 0.11Dc | 5.90 ± 0.41Cab | 11.11 ± 0.04Aa | 41.56 ± 0.52Bb | 49.13 ± 1.73Bb |
| M2 | 0–5 | 0.65 ± 0.02Aa | 0.63 ± 0.03Ab | 0.50 ± 0.02Ab | 0.63 ± 0.02Ab | 17.73 ± 0.16Bb | 17.95 ± 0.29Ba | 43.80 ± 0.93Ab | 7.38 ± 0.05Cd | 8.23 ± 0.27Ab | 8.10 ± 0.04Db | 88.22 ± 0.61Ac | 42.13 ± 1.47Bd |
| | 5–10 | 0.54 ± 0.01Ba | 0.56 ± 0.02Bb | 0.42 ± 0.01Bc | 0.48 ± 0.02Cb | 18.06 ± 0.17ABab | 18.33 ± 0.21Aa | 39.70 ± 0.59Bb | 10.18 ± 0.30Ab | 7.37 ± 0.20Bb | 10.65 ± 0.04Ab | 69.13 ± 0.69Bc | 42.75 ± 1.49ABd |
| | 10–20 | 0.42 ± 0.02Cb | 0.46 ± 0.02Ca | 0.33 ± 0.01Cc | 0.43 ± 0.01Dc | 18.44 ± 0.26Aa | 17.65 ± 0.02Ba | 20.71 ± 0.70Cb | 9.46 ± 0.29Bb | 7.57 ± 0.38Ba | 9.70 ± 0.25Bc | 42.09 ± 0.35Cc | 42.35 ± 1.47Bc |
| | 20–40 | 0.41 ± 0.01Ca | 0.35 ± 0.01Dc | 0.28 ± 0.02Dc | 0.53 ± 0.01Bc | 17.77 ± 0.25Ba | 15.25 ± 0.12Cb | 16.80 ± 0.99Db | 9.59 ± 0.17Bb | 6.10 ± 0.32Cab | 8.47 ± 0.14Cc | 41.56 ± 0.35Cb | 45.40 ± 1.58Ac |
| M3 | 0–5 | 0.62 ± 0.02Aa | 0.58 ± 0.02Aa | 0.52 ± 0.01Cb | 0.47 ± 0.02ABd | 18.17 ± 0.36Aab | 15.80 ± 0.31Ac | 47.85 ± 1.19Ac | 10.04 ± 0.24Ac | 7.33 ± 0.39Ac | 8.47 ± 0.28Ab | 103.12 ± 1.39Aa | 88.21 ± 3.11Aa |
| | 5–10 | 0.55 ± 0.01Ba | 0.52 ± 0.02Bc | 0.63 ± 0.01Ab | 0.44 ± 0.02Bc | 18.38 ± 0.17Aa | 15.78 ± 0.34Ac | 42.15 ± 1.28Ba | 9.10 ± 0.28Bc | 6.67 ± 0.47Bc | 8.13 ± 0.12Bc | 102.10 ± 1.32Aa | 69.13 ± 2.42Ba |
| | 10–20 | 0.47 ± 0.01Ca | 0.44 ± 0.02Cab | 0.57 ± 0.02Bb | 0.49 ± 0.02Ab | 17.95 ± 0.15ABb | 15.71 ± 0.20Ab | 43.26 ± 0.77Ba | 8.47 ± 0.08Cc | 5.67 ± 0.27Cb | 5.93 ± 0.04Cd | 65.67 ± 0.75Bb | 42.10 ± 1.47Cc |
| | 20–40 | 0.42 ± 0.02Da | 0.43 ± 0.02Cb | 0.58 ± 0.02Bb | 0.37 ± 0.01Cd | 17.60 ± 0.19Ba | 15.92 ± 0.23Aa | 19.68 ± 0.83Ca | 5.89 ± 0.11Dd | 5.70 ± 0.10Cb | 5.68 ± 0.05Cd | 58.60 ± 0.72Ca | 41.56 ± 1.45Cd |
| CK | 0–5 | 0.43 ± 0.02Bb | 0.41 ± 0.02Ad | 0.74 ± 0.03Aa | 0.75 ± 0.03Aa | 18.48 ± 0.14Aa | 15.14 ± 0.26Bd | 21.26 ± 0.91Ad | 13.01 ± 0.32Aa | 10.60 ± 0.40Aa | 10.61 ± 0.30Ba | 97.56 ± 1.33Ab | 79.01 ± 2.75Ab |
| | 5–10 | 0.39 ± 0.01Ac | 0.39 ± 0.02Ad | 0.62 ± 0.01Cb | 0.65 ± 0.02Ba | 18.2 ± 0.24ABa | 16.33 ± 0.31Ab | 16.52 ± 0.53Bc | 11.26 ± 0.14Ba | 8.00 ± 0.24Ba | 11.12 ± 0.28Aa | 84.25 ± 0.81Bb | 59.62 ± 2.08BCb |
| | 10–20 | 0.41 ± 0.01Bb | 0.42 ± 0.02Ab | 0.70 ± 0.01Ba | 0.69 ± 0.02Ba | 17.91 ± 0.19BCb | 15.84 ± 0.38Ab | 14.65 ± 0.73Cd | 10.66 ± 0.19Ca | 7.10 ± 0.26Ca | 11.20 ± 0.18Ab | 69.89 ± 0.62Ca | 57.49 ± 2.03Ca |
| | 20–40 | 0.32 ± 0.01Cb | 0.35 ± 0.01Bc | 0.71 ± 0.02ABa | 0.79 ± 0.03Aa | 17.73 ± 0.10Ca | 15.01 ± 0.29Bb | 13.25 ± 0.92Cc | 10.40 ± 0.20Ca | 6.30 ± 0.13Da | 10.29 ± 0.15Bb | 59.72 ± 0.87Da | 63.80 ± 2.22Ba |
| Treatments | | *** | *** | *** | *** | *** | *** | *** | *** | *** | *** | *** | *** |
| Soil layers | | *** | *** | *** | *** | *** | *** | *** | *** | *** | *** | *** | *** |
| Treatments× Soil layers | | *** | *** | *** | *** | *** | *** | *** | *** | *** | *** | *** | *** |

**Note:**

M1: Lingwu jujube-low-density perennial ryegrass intercropping; M2: Lingwu jujube-medium-density perennial ryegrass intercropping; M3: Lingwu jujube-high-density perennial ryegrass intercropping; CK indicates Lingwu jujube monocropping; different upper-case letters in the same column indicate that the difference between different soil layers under the same density is significant ($p < 0.05$); different lower-case letters in the same column indicate that the difference between the same soil layers under the different density is significant ($p < 0.05$). *, **, ***, **** and ns represent two-way ANOVA $p < 0.05$, $p < 0.01$, $p < 0.001$ and not significant, respectively (Values are means of three repetitions ± standard error).

**Table 5 Changes in soil organic matter, pH and cation exchange under date palm canopy in different treatments.**

| Treatments | Soil layers (cm) | SOC (g kg⁻¹) | | pH- | | CEC (cmol+ kg⁻¹) | |
|---|---|---|---|---|---|---|---|
| | | 2023 | 2024 | 2023 | 2024 | 2023 | 2024 |
| M1 | 0–5 | 10.16 ± 0.31Ab | 10.65 ± 0.30Ac | 8.22 ± 0.18Ab | 7.97 ± 0.28Ac | 12.18 ± 0.37Ab | 12.47 ± 0.26Ab |
| | 5–10 | 8.56 ± 0.22Bc | 8.19 ± 0.33Cb | 8.13 ± 0.20Aa | 7.87 ± 0.20Ab | 10.69 ± 0.32Bb | 10.90 ± 0.24Bb |
| | 10–20 | 7.98 ± 0.18Cc | 9.01 ± 0.36Bc | 8.14 ± 0.09Aab | 7.91 ± 0.23Ab | 10.98 ± 0.33Bb | 11.23 ± 0.25Bb |
| | 20–40 | 4.01 ± 0.27Dc | 5.70 ± 0.22Dc | 8.16 ± 0.13Aa | 7.91 ± 0.19Aa | 9.40 ± 0.28Cb | 9.60 ± 0.25Cb |
| M2 | 0–5 | 13.24 ± 0.19Aa | 13.40 ± 0.54Aa | 8.10 ± 0.04ABb | 8.09 ± 0.28Ab | 13.25 ± 0.40Aa | 13.58 ± 0.35Aa |
| | 5–10 | 11.65 ± 0.21Ba | 11.77 ± 0.47BCa | 7.88 ± 0.24Ba | 7.93 ± 0.38Ab | 12.98 ± 0.39Aa | 13.32 ± 0.37Aa |
| | 10–20 | 11.78 ± 0.20Bb | 11.95 ± 0.48Bb | 8.17 ± 0.09Aab | 8.34 ± 0.29Aa | 13.01 ± 0.39Aa | 13.29 ± 0.38Aa |
| | 20–40 | 10.89 ± 0.13Cb | 10.99 ± 0.44Cb | 8.14 ± 0.1ABa | 8.13 ± 0.24Aa | 10.56 ± 0.32Ba | 10.84 ± 0.31Ba |
| M3 | 0–5 | 10.11 ± 0.74Cb | 12.53 ± 0.50BCb | 8.14 ± 0.15Ab | 8.32 ± 0.29Ab | 11.15 ± 0.33Ac | 11.42 ± 0.3Ac |
| | 5–10 | 11.08 ± 0.45Cb | 11.60 ± 0.46Ca | 8.12 ± 0.23Aa | 7.91 ± 0.31Ab | 10.35 ± 0.31Bb | 10.60 ± 0.29Bb |
| | 10–20 | 14.32 ± 0.48Aa | 14.70 ± 0.58Aa | 8.02 ± 0.10Ab | 8.18 ± 0.29Aa | 9.48 ± 0.29Cc | 9.68 ± 0.27Cc |
| | 20–40 | 12.93 ± 0.75Ba | 13.25 ± 0.53Ba | 8.00 ± 0.09Aa | 8.18 ± 0.20Ab | 9.40 ± 0.28Cb | 9.63 ± 0.26Cb |
| CK | 0–5 | 9.12 ± 0.13Ac | 9.26 ± 0.37Ad | 8.64 ± 0.38Aa | 8.47 ± 0.30Aa | 8.79 ± 0.27ABd | 8.98 ± 0.26ABd |
| | 5–10 | 9.01 ± 0.10Ac | 7.91 ± 0.31Bb | 8.33 ± 0.31Aa | 8.20 ± 0.27Aa | 8.50 ± 0.26Bc | 8.68 ± 0.24Bc |
| | 10–20 | 7.59 ± 0.19Bc | 7.81 ± 0.31Bd | 8.32 ± 0.17Aa | 8.43 ± 0.19Aa | 8.37 ± 0.25Bd | 8.57 ± 0.23Bd |
| | 20–40 | 4.65 ± 0.12Cc | 3.67 ± 0.15Cd | 8.17 ± 0.12Aa | 8.31 ± 0.29Aa | 9.12 ± 0.27Ab | 9.32 ± 0.28Ab |
| Treatments | | *** | *** | ** | * | *** | *** |
| Soil layers | | *** | *** | ns | ns | *** | *** |
| Treatments × Soil layers | | *** | *** | ns | ns | *** | *** |

Note:
M1: Lingwu jujube-low-density perennial ryegrass intercropping; M2: Lingwu jujube-medium-density perennial ryegrass intercropping; M3: Lingwu jujube-high-density perennial ryegrass intercropping; CK indicates Lingwu jujube monocropping; different upper-case letters in the same column indicate that the difference between different soil layers under the same density is significant ($p < 0.05$); different lower-case letters in the same column indicate that the difference between the same soil layers under the different density is significant ($p < 0.05$). *, **, *** and ns represent two-way ANOVA $p < 0.05$, $p < 0.01$, $p < 0.001$ and not significant, respectively (Values are means of three repetitions ± standard error).

layer over 2 years was significantly higher in the high-density ryegrass and 'Lingwu Changzao' jujube intercropping system than in the low- and medium-density systems and the monoculture system, with increases of 35.58–36.60%, 1.8–7.6%, and 37.3–49.99%, respectively. In specific soil layers, except for the high-density system, all other systems showed a decreasing trend in soil organic matter content with increasing soil depth. Soil pH significantly influences soil properties and fertility, thereby directly or indirectly affecting plant growth. During the experimental period, soil pH in the 0–40 cm layer was significantly lower in all three ryegrass and jujube intercropping systems compared to the monoculture system. The low-density intercropping system had a pH reduction of 2.44–5.23%, the medium-density system 2.75–3.50%, and the high-density system 2.45–3.54% compared to the monoculture. Among different soil layers, the lowest pH in 2023 was observed in the 5–10 cm layer of the medium-density system, with a value of 7.88. In 2024, the lowest pH values were found in the 10–20 cm and 20–40 cm layers of the low-density system, both at 7.91. The highest soil pH over the 2 years was observed in the 0–5 cm layer of the monoculture system, with a value of 8.64. Soil cation exchange capacity

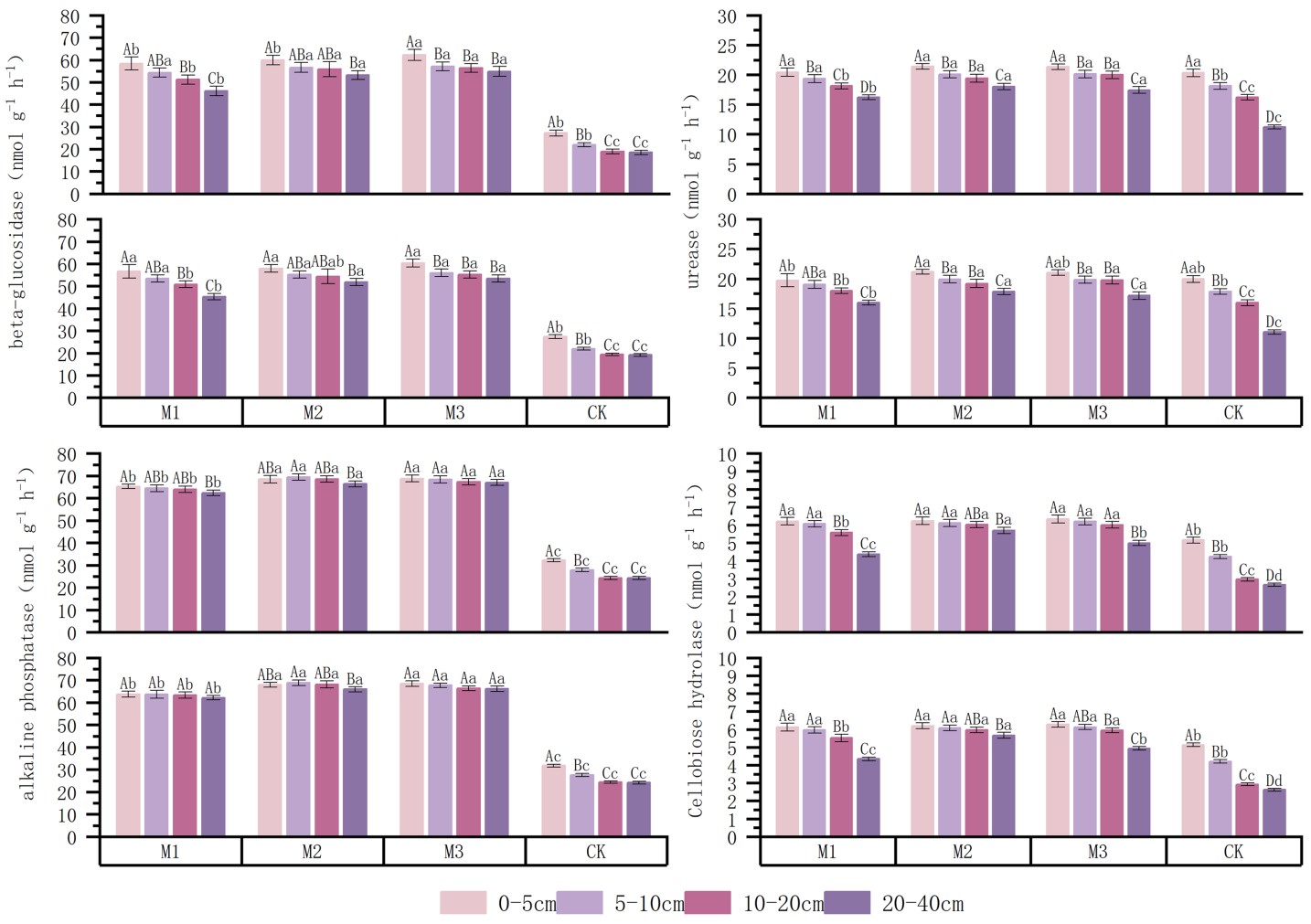

**Figure 6 Enzyme activities related to carbon and nitrogen cycling in soil under different cropping patterns, 2023–2024.** M1: Lingwu jujube-low-density perennial ryegrass intercropping; M2: Lingwu jujube-medium-density perennial ryegrass intercropping; M3: Lingwu jujube-high-density perennial ryegrass intercropping; CK indicates Lingwu jujube monocropping; different upper-case letters in the same column indicate that the difference between different soil layers under the same density is significant ($p < 0.05$); different lower-case letters in the same column indicate that the difference between the same soil layers under the different density is significant ($p < 0.05$). (Values are means of three repetitions ± standard error).

(CEC) is directly related to nutrient retention ability and nutrient supply to plants. Different ryegrass intercropping densities and soil depths significantly affected soil CEC ($p < 0.001$). Intercropping increased CEC across all soil layers. Compared to the monoculture system, the low-, medium-, and high-density intercropping systems showed CEC increases of 19.55–19.57%, 30.16–30.31%, and 13.86–14.02%, respectively. In all soil layers, the low-, medium-, and high-density intercropping systems exhibited a decreasing trend in CEC with increasing soil depth.

### Soil enzyme activity beneath the jujube tree canopy

As shown in Fig. 6, compared to the monoculture system, the low-, medium-, and high-density ryegrass and 'Lingwu Changzao' jujube intercropping systems increased alkaline phosphatase activity by 57.39–57.51%, 59.13–60.20%, and 55.54–59.90%,

respectively. Intercropping significantly enhanced soil alkaline phosphatase activity, with the medium-density system showing a greater increase than the other two densities. Across all soil layers, alkaline phosphatase activity decreased with increasing soil depth. Urease is a specific enzyme that catalyzes the hydrolysis of urea into ammonia, carbon dioxide, and water. Urease activity is strongly influenced by soil moisture. Optimal moisture enhances urease activity, but excessive moisture can reduce activity due to insufficient oxygen. The figure indicates that intercropping increased urease activity in the soil, though this increase was less pronounced in the shallow soil layers. In the 0–40 cm soil layer, the low-, medium-, and high-density intercropping systems increased urease activity by 10.70–11.94%, 16.56–17.03%, and 16.45–16.98%, respectively, compared to the monoculture system. The most significant increases were observed at the 20–40 cm soil depth, with increases of 30.72–30.96%, 37.65–38.57%, and 32.53–35.70%, respectively. However, urease activity decreased with increasing soil depth. Low soil moisture reduces microbial metabolic activity, leading to decreased β-glucosidase activity. Prolonged drought significantly decreases soil enzyme activity, affecting the decomposition of organic matter and nutrient cycling. All three intercropping densities significantly increased β-glucosidase activity, with the medium- and high-density intercropping systems showing the greatest increases, ranging from 59.99–61.54% and 60.97–62.36%, respectively. Across different soil layers, β-glucosidase activity was higher in the high-density intercropping system than in the other treatments. Cellulase activity beneath the jujube tree canopy was significantly enhanced by the low-, medium-, and high-density intercropping systems, with increases of 30.27–32.45%, 30.16–37.63%, and 13.86–36.23%, respectively, compared to the monoculture system. Among the three densities, the medium-density intercropping system exhibited the highest average cellulase activity in the 0–40 cm soil layer, 7.66–8.19% higher than in the low-density system and 2.19–2.58% higher than in the high-density system. Across all soil layers, cellulase activity decreased with increasing soil depth.

### Changes in soil moisture content across different soil layers beneath the jujube tree canopy

Figure 7 illustrates changes in soil moisture content within the 0–180 cm soil layer beneath the jujube tree canopy. The 0–20 cm layer is primarily influenced by evaporation, as it is affected by both plant water uptake and evaporation. Over the 2-year experiment, compared to the monoculture system, soil moisture content in the 0–20 cm range increased by 39.32–41.03%, 26.60–30.51%, and 11.63–24.18% in the low-, medium-, and high-density ryegrass and 'Lingwu Changzao' jujube intercropping systems, respectively. In the low-density intercropping system, soil moisture content decreased with increasing soil depth, particularly at the 60–180 cm depth, where it was the lowest (6.30%). In the medium-density intercropping system, a similar trend was observed, with soil moisture content decreasing with depth. However, the reduction was more pronounced at the 60–180 cm depth (4.40%). Compared to 2023, soil moisture content in 2024 increased across all depths, particularly at the 60–180 cm depth, where it rose to 8.13%. In the high-density ryegrass and jujube intercropping system, soil moisture content in the 0–20 cm layer was slightly higher than in the medium-density system. However, moisture

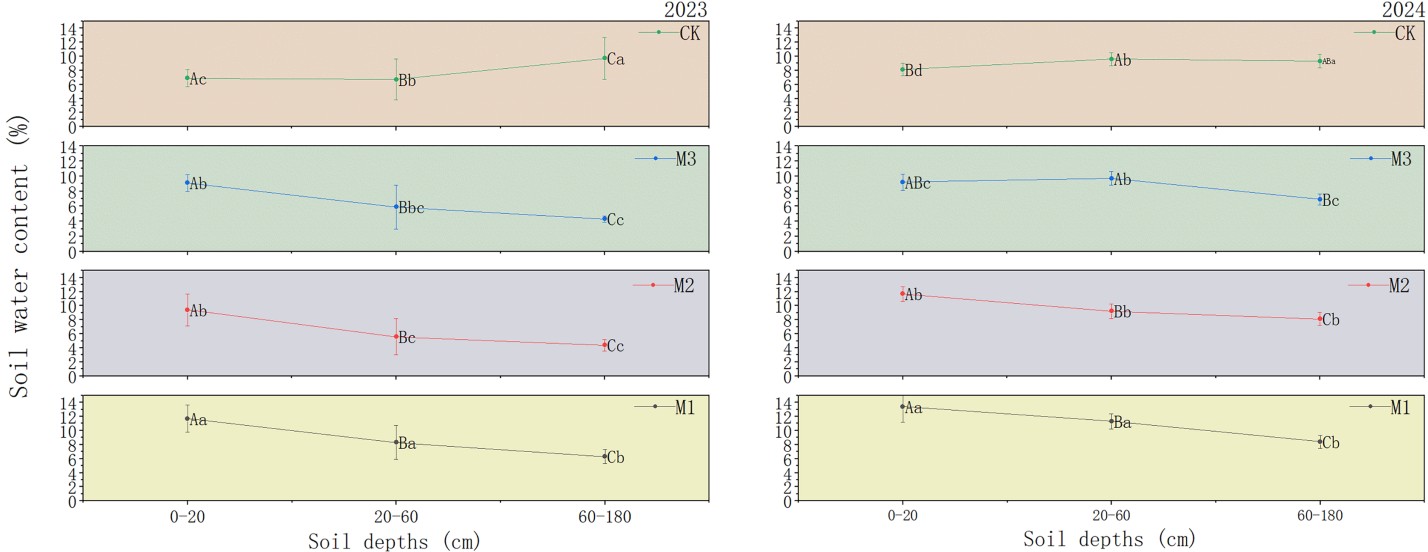

**Figure 7 Soil moisture content from 0–180 cm under different cropping patterns, 2023–2024.** M1: Lingwu jujube-low-density perennial ryegrass intercropping; M2: Lingwu jujube-medium-density perennial ryegrass intercropping; M3: Lingwu jujube-high-density perennial ryegrass intercropping; CK indicates Lingwu jujube monocropping; different upper-case letters in the same column indicate that the difference between different soil layers under the same density is significant ($p < 0.05$); different lower-case letters in the same column indicate that the difference between the same soil layers under the different density is significant ($p < 0.05$). (Values are means of three repetitions ± standard error).

content at the 20–60 cm and 60–180 cm depths remained relatively stable but showed a gradual decreasing trend.

### Correlation analysis between soil moisture, soil enzyme activity, and soil physical properties in the 0–180 cm layer beneath the jujube tree canopy

This study investigates the 0–180 cm soil layer, focusing on the interactions among soil moisture, physical properties, and enzyme activity (Fig. 8). Analyzing these correlations is crucial to elucidating the processes and mechanisms that influence water-soil properties and enzyme activity in an intercropping system. Pearson correlation analysis from 2023 reveals that in the 0–20 cm layer, organic matter, cellobiohydrolase, and soil moisture are strongly positively correlated ($p < 0.01$). Conversely, the soil's mean weight diameter shows a significant negative correlation ($p < 0.01$). In the 0–60 cm layer, both field water capacity and total soil porosity significantly positively correlate with soil moisture ($p < 0.01$). Soil moisture in the 60–180 cm layer is significantly negatively correlated with soil bulk density ($p < 0.01$). There is a negative correlation between soil pH, β-glucosidase, and urease ($p < 0.05$). In 2024, strong positive correlations are observed between soil moisture and variables such as field water capacity, organic matter, alkaline phosphatase, soil aggregate stability, and geometric mean diameter in the 0–20 cm layer ($p < 0.01$); significant negative correlations with fractal dimension are also noted. A significant positive correlation with available phosphorus is noted in the 20–60 cm layer ($p < 0.01$). Although a negative correlation trend is suggested for soil aggregate stability, it does not reach significance. In the 60–180 cm layer, significant positive correlations are exhibited between soil moisture and both total phosphorus and alkali-hydrolyzable nitrogen ($p < 0.01$); a weaker,

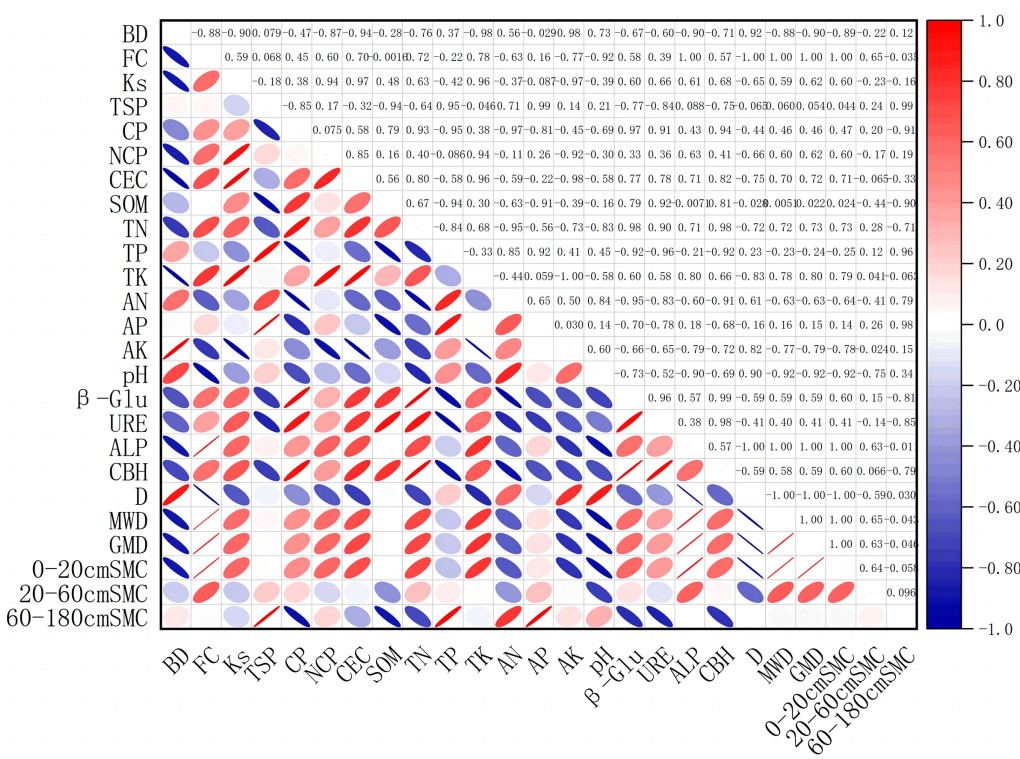

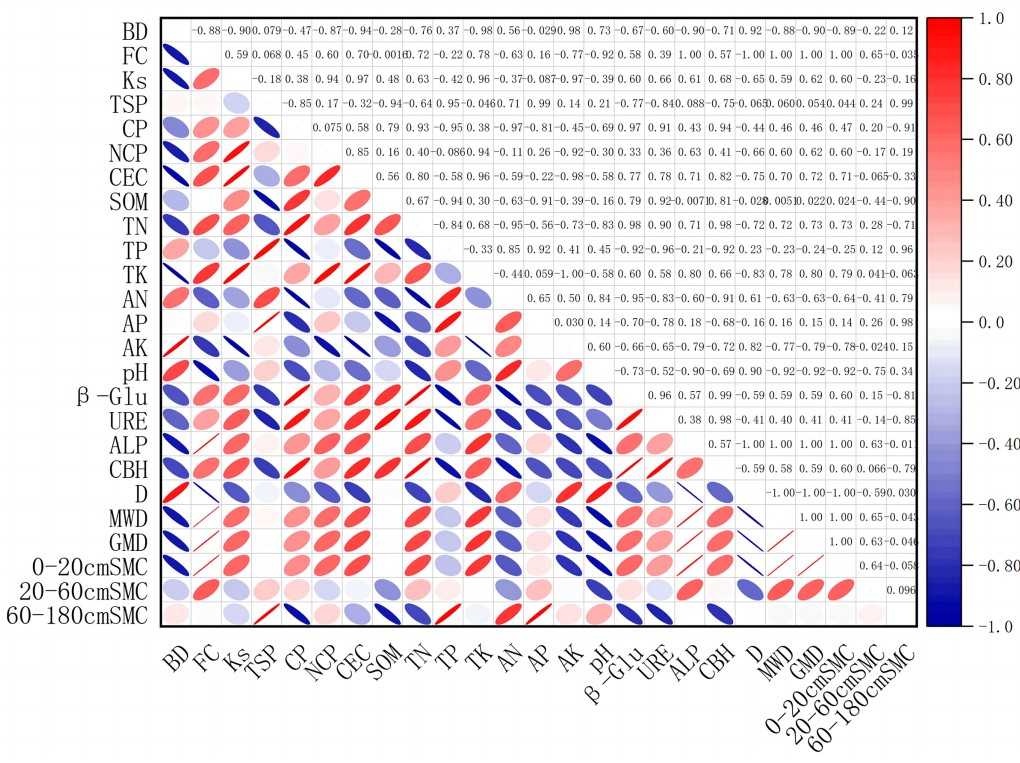

**Figure 8 Correlation analysis of soil moisture (0–180 cm), soil enzyme activities, and soil physical properties.** BD, Bulk Density; FC, Field Capacity; Ks, Saturated Hydraulic Conductivity; TSP, Total Soil Porosity, CP, Capillary Porosity, NCP, Non-Capillary Porosity; CEC, Cation Exchange Capacity;

**Figure 8** (continued)
SOM, Soil Organic Matter; TN, Total Nitrogen; TP, Total Phosphorus; TK, Total Potassium; AN, Alkaline Hydrolyzed Nitrogen; AP, Available Phosphorus; AK, Available Potassium; pH, Soil pH; β-Glu, β-Glucosidase; URE, Urease; ALP, Alkaline Phosphatase; CBH, Cellobiohydrolase; D, Fractal Dimension; MWD, Mean Weight Diameter; GMD, Geometric Mean Diameter; SMC, Soil Moisture.

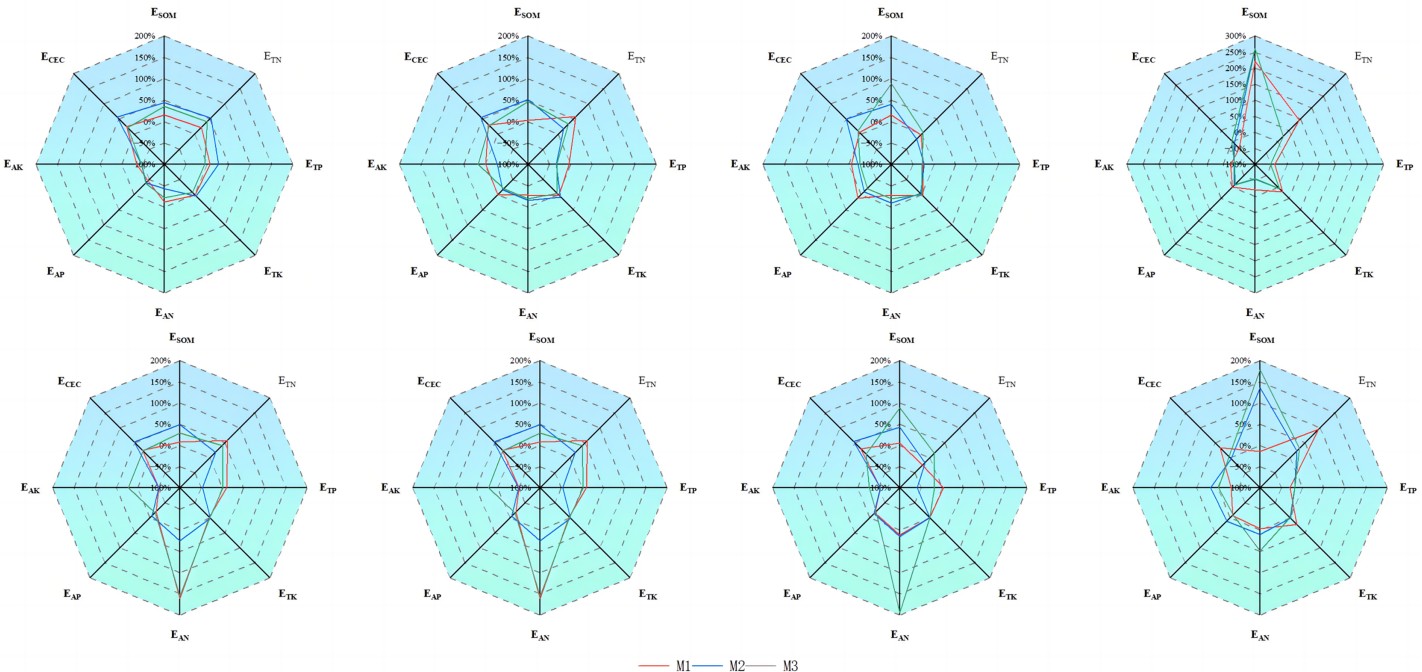

**Figure 9 Soil nutrient effects under the canopy of date palm in different treatments.** ECEC-soil cation exchange capacity effect, ESOM-organic matter effect, ETN-whole nitrogen effect, ETP-whole phosphorus effect, ETK-whole potassium effect, EAN-alkaline dissolved nitrogen effect, EAP-effective phosphorus effect, EAK-rapid potassium effect.

nonsignificant negative correlation with organic matter is also observed. A significant negative correlation with urease is observed ($p < 0.01$).

### Soil nutrient effects beneath the jujube tree canopy under different treatments

In the low-density ryegrass and 'Lingwu Changzao' jujube intercropping system, the effects of soil nutrients in different soil layers varied compared to the monoculture system, with some similarities observed over the 2 years (Fig. 9). Organic matter had a positive effect in the 0–5 cm, 5–10 cm, and 10–20 cm layers, but a negative effect in the 20–40 cm layer. Available potassium showed a negative effect across all soil layers; however, the magnitude of this negative effect decreased with increasing soil depth. The positive effect of alkaline-hydrolyzable nitrogen was most pronounced in the 0–5 cm layer, reaching 171.40%. In the medium-density ryegrass and jujube intercropping system, both organic matter and alkaline-hydrolyzable nitrogen had positive effects across all soil layers. The positive effect of organic matter increased with soil depth, reaching 136.13% in the 20–40 cm layer. In contrast, the positive effect of alkaline-hydrolyzable nitrogen gradually

decreased with soil depth, reaching 106.20% in the 0–5 cm layer. Similarly, in the high-density ryegrass and jujube intercropping system, organic matter continued to have positive effects across all soil layers. Alkaline-hydrolyzable nitrogen had positive effects across all soil layers in 2023 but showed negative effects in 2024. Available phosphorus showed negative effects in all soil layers except the 0–5 cm layer. Soil cation exchange capacity showed positive effects in all layers except the 0–5 cm layer. Overall, the low-density intercropping system performed relatively well in the shallow soil layers (0–10 cm), with improvements observed in 2024 compared to 2023. The medium-density intercropping system performed better across most soil layers and indicators, particularly in terms of organic matter and total nitrogen efficiency, demonstrating a greater soil improvement effect.

## DISCUSSION

### Effects of different densities of ryegrass and 'Lingwu Changzao' jujube intercropping systems on soil aggregates beneath the jujube tree canopy

Soil aggregate stability is a critical characteristic that influences soil sustainability and crop production (*Amézketa, 1999*). In this study, soil water-stable aggregates under different densities of ryegrass and 'Lingwu Changzao' jujube intercropping systems were predominantly microaggregates (<0.25 mm). Compared to the monoculture of 'Lingwu Changzao' jujube, the intercropping systems with different ryegrass densities increased the content of macroaggregates (>0.25 mm). This finding is consistent with the results of *Chen et al. (2017)* who demonstrated that after 10 years, intercropping rubber trees with four other crops significantly increased the content of soil macroaggregates in rubber plantations. The likely reason for this is that intercropping systems often promote the accumulation of organic matter in the soil through permanent vegetation cover, which enhances soil microbial populations and increases the content of soil macroaggregates. Additionally, intercropped plants may induce compression by their fine roots, further contributing to the increase in soil macroaggregate content. Our results are also consistent with the findings of *Tao et al. (2021)*.

### Effects of different densities of ryegrass and 'Lingwu Changzao' jujube intercropping systems on soil bulk density, field water holding capacity, saturated hydraulic conductivity, and soil porosity beneath the jujube tree canopy

Soil physical properties significantly influence aeration, infiltration, water retention, solute transport, and erosion resistance (*Liu et al., 2022*). Undoubtedly, intercropping systems can alter soil bulk density (*Duan et al., 2024*), field water-holding capacity, saturated hydraulic conductivity (*Secco et al., 2023*), and porosity to varying degrees. Our results indicate that different densities of ryegrass and 'Lingwu Changzao' jujube intercropping systems had a limited impact on soil bulk density beneath the jujube canopy, consistent with the findings of *Cardinael et al. (2015)*. In this study, soil bulk density under

intercropping systems tended to decrease compared to the monoculture system. This could be attributed to the well-developed root system of ryegrass, which enhances soil looseness, reducing bulk density. Lower bulk density generally indicates looser soil, facilitating root growth and water infiltration. This effect was more pronounced in the shallow soil layers, as corroborated by subsequent soil moisture findings. Our results align with those of *Qiang (2020)*. Generally, as ryegrass density increases, soil field water-holding capacity also increases, as denser vegetation reduces surface runoff and enhances soil water retention. However, changes in saturated hydraulic conductivity depend on the extent of soil structure improvement. Although intercropping increased soil field water-holding capacity beneath the jujube canopy, the trend did not correlate with crop density, possibly because ryegrass root distribution became more concentrated at medium and high densities rather than extending throughout the soil profile. In such cases, the improvement in soil structure by roots might reach saturation, leading to a less significant change in water-holding capacity. The medium-density ryegrass and 'Lingwu Changzao' jujube intercropping system exhibited the best field water-holding capacity beneath the jujube canopy, possibly because medium-density ryegrass managed soil water dynamics more effectively, preventing excessive water evaporation and runoff, thus maintaining soil moisture. This stable moisture condition helps sustain and promote soil structure improvement, leading to increased saturated hydraulic conductivity.

## Effects of different densities of ryegrass and 'Lingwu Changzao' jujube intercropping systems on soil nutrients, pH, and enzyme activity beneath the jujube tree canopy

In this study, different densities of ryegrass and 'Lingwu Changzao' jujube intercropping systems increased the content of total nitrogen, total potassium, available phosphorus, and available potassium in the shallow soil layers (0–5 cm, 5–10 cm) beneath the jujube canopy, consistent with the findings of *Gabhane et al. (2023)*, *Ma et al. (2017)*, and *Bhagat et al. (2024)* found that intercropping pear with ten different crops was highly effective in improving overall soil quality, including physical, chemical, and biological indicators. We believe that the increase in soil phosphorus availability can be attributed to the intercropping system's ability to increase the microbial community in the rhizosphere, particularly phosphorus-solubilizing microbes. Compared to other intercropped crops, ryegrass has a shallower root system, enabling it to positively influence phosphorus, typically concentrated in the upper soil layers, thereby enhancing its availability (*Wenjing et al., 2024*). *Swain (2016)* indicated that intercropping guava with legumes could enhance soil nitrogen levels. However, studies by *Zhu et al. (2022)* and *Wang et al. (2015)* suggested that the improvement in soil nitrogen is not limited to leguminous crops. the enhancement of soil nutrients by ryegrass intercropping varied significantly with different densities, mainly reflected in the positive and negative effects across different soil layers and years. Ryegrass and 'Lingwu Changzao' jujube have different nutrient demands and absorption rates. Ryegrass generally has a robust root system, allowing it to rapidly absorb nutrients in the shallow soil, while 'Lingwu Changzao' jujube may rely more on nutrients from deeper

soil layers. This difference in root distribution and nutrient absorption may lead to varying nutrient depletion rates in different soil layers, resulting in differential soil nutrient effects (*Lilan et al., 2024*). The differing performance of organic matter in 2023 and 2024 may be due to the ability of ryegrass roots in the intercropping system to promote organic matter accumulation, particularly in the shallow soil layers. This accumulation could improve soil structure, enhancing water retention and nutrient supply. However, in the early stages (first year), this accumulation might not have been significant enough, and nutrient competition could have led to depletion, resulting in a "negative enhancement" effect. Over time (second year), organic matter accumulation and soil structure improvement became apparent, leading to improved soil nutrient status, manifested as "positive enhancement". Changes in pH can significantly affect soil nutrient dynamics and crop growth. Our experimental results showed that the intercropping system did not significantly improve pH, especially in the first year of the experiment, where no significant differences were observed in pH across treatments and soil layers. However, in the second year, all three intercropping densities reduced soil pH beneath the jujube canopy to varying degrees. The change in pH is likely related to the choice of intercropped crops (*Omondi & Kniss, 2014*). Hypothesized that intercropping ryegrass, wheat, oats, and maize could alleviate iron deficiency in dry beans but found no significant difference in pH between treatments. Our experimental results are similar to those of *Lu et al. (2024)*, where the intercropping system brought soil pH closer to neutral. pH significantly affects soil enzyme activity. showed that orchard intercropping increased soil enzyme activity, and that enzyme activity (β-glucosidase, urease, alkaline phosphatase) was negatively correlated with soil pH. *Donald A'Bear et al. (2014)* indicated that β-glucosidase was significantly positively correlated with total nitrogen, a conclusion also supported by our findings. Regarding the relationship between enzyme activity and soil depth, our results show that soil enzyme activity decreases with increasing soil depth, consistent with the findings of *Xuan (2024)*.

### Effects of different densities of ryegrass and 'Lingwu Changzao' jujube intercropping systems on soil moisture content beneath the jujube tree canopy

Intercropping significantly increased soil moisture content in the 0–20 cm and 20–60 cm soil layers. We believe that the fine root system of ryegrass contributes to increased soil porosity, promoting better soil structure formation and enhancing soil moisture retention capacity. Additionally, planting ryegrass between orchard rows can enhance surface cover, reduce ineffective surface water evaporation, and effectively intercept rainfall and surface runoff, thereby improving soil moisture retention (*Xianglong et al., 2024*). Our results are consistent with the conclusions of *Palese et al. (2014)*. *Wei et al. (2024)* intercropped ryegrass with apple trees and found that soil moisture content in each layer of the apple/ryegrass intercropping system was generally higher than in the apple monoculture system. Our results indicate that soil moisture content in the 60–180 cm soil layer under the monoculture system was significantly higher than in the intercropping system. We believe this is partly because the intercropping system may more effectively retain surface runoff,

allowing moisture to be utilized or evaporated in the shallower soil layers rather than percolating into deeper layers. On the other hand, jujube trees in the monoculture system may have deeper and more widespread root systems, enabling them to utilize deep soil moisture more effectively. In contrast, in the intercropping system, ryegrass may compete for moisture in the shallower soil layers, affecting the replenishment and conservation of deep soil moisture (*Liang et al., 2024*). Additionally, correlation studies show that within the 0–180 cm soil layer, soil moisture and physical properties are closely related to enzyme activity, aligning with the findings of *Qiang (2020)*, where soil moisture exhibited a strong correlation with bulk density, soil porosity, and other indicators.

## CONCLUSIONS

The cultivation of Lingwu jujube traditionally employs clean tillage, leaving substantial gaps between rows and exposing almost 60% of the orchard to the elements. This method promotes rapid soil moisture evaporation, exacerbates soil erosion, and deteriorates the soil's physicochemical properties. Consequently, there is a critical need for a more sustainable planting approach that optimally utilizes land resources. A pertinent question is whether varying densities of ryegrass intercropping can improve the uptake of nutrients and water by the jujube tree, the primary species in this ecosystem. In this context, a 2-year field experiment was conducted with three densities of perennial ryegrass intercropped with Lingwu jujube. The experiment assessed the impact on soil's physical and chemical attributes beneath the jujube canopy, with a focus on correlating soil moisture, enzyme activity, and physical properties. The findings reveal that intercropping at a medium density most effectively enhanced the soil's physical characteristics. Relative to monoculture, this approach increased the proportion of water-stable aggregates (0.5–0.25 mm) by 4.16%, decreased the soil's fractal dimension by 0.46%, augmented the field water holding capacity by 14.78%, and significantly boosted soil enzyme activity. Furthermore, high-density ryegrass intercropping elevated the soil's organic matter content by 36.09% and ameliorated both the pH and cation exchange capacity. Conversely, low-density intercropping raised soil moisture levels by 40.18% in the top 20 cm of the soil. Collectively, these results suggest that an optimal density of ryegrass in intercropping not only bolsters the moisture retention capabilities of soil in Lingwu jujube orchards but also enhances overall soil fertility. Therefore, the adoption of ryegrass and jujube tree intercropping is highly advisable in the ecologically sensitive and resource-constrained arid sandy regions of northern China, offering substantial practical benefits.

### Funding

This work was supported by the Key Research & Development Project of Ningxia Autonomous Region, China (grant No. 2023BEG02039). The funders had no role in study design, data collection and analysis, decision to publish, or preparation of the manuscript.

## Grant Disclosures

The following grant information was disclosed by the authors:
Key Research & Development Project of Ningxia Autonomous Region, China:
2023BEG02039.

## Competing Interests

The authors declare that they have no competing interests.

## Author Contributions

- Yao Ma conceived and designed the experiments, performed the experiments, prepared figures and/or tables, and approved the final draft.
- Bin Cao conceived and designed the experiments, analyzed the data, prepared figures and/or tables, authored or reviewed drafts of the article, and approved the final draft.
- Xiaojia Wang performed the experiments, analyzed the data, prepared figures and/or tables, and approved the final draft.
- Weijun Chen performed the experiments, prepared figures and/or tables, and approved the final draft.

## Data Availability

The raw measurements are available in the Supplemental File.

## Supplemental Information

Supplemental information for this article can be found online at http://dx.doi.org/10.7717/peerj.18710#supplemental-information.

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
