# Peer review of "Effect of intercropping Lolium perenne in Ziziphus jujuba orchards on soil quality in the canopy"

_PeerJ, doi:10.7717/peerj.18710_

## Round 0.1 · original submission · Major Revisions

I ask you to correct the title of the article: "Effect of dryland ryegrass on soil quality under Ziziphus jujuba canopy". In this case, it is also better to replace ryegrass with the Latin name of the genus or a specific plant species.

I ask you to adapt the manuscript to modern methods of statistical data processing. The data in Figures 3-9, 12, 13 should be presented in the form of a box analysis (median, first and third quartiles, minimum and maximum values). This will increase readers' confidence in the published data. In the titles of tables and figures, the replication of studies should be indicated in parentheses. In the titles of tables, the standard error or standard deviation indicated in the table should be indicated in parentheses. Table 3 requires statistical processing (one-time changes cannot be published).

The bibliography does not meet the requirements of the journal. All % in the text of the article should be rounded equally for each of the characteristics. It is not correct to write "(p < 0.05)" or "(p > 0.05)" in the text of the article: it is desirable to indicate a specific value of the reliability of differences.

The text of the article contains a large number of technical errors that can be corrected if the authors provide the manuscript for analysis to an experienced technical scientific editor.

In addition, I ask you to carefully correct the manuscript in accordance with each of the reviewers' comments.

I hope that the quality of the improvement of this manuscript will allow it to be published in the journal and will not require repeated in-depth review.

Reviewer 1 ·

Basic reporting

This study analyzed the influences of ryegrass-jujube intercropping system on in the different densities of ryegrass-planting, which provides theoretical support for promoting this intercropping system in resource-limited andecologically fragile semi-arid and arid regions of northern China. However, in the section of ABSTRACT and INTRODUCTION, the purpose of this study is not prominent, and it needs to be further condensed, after all, cropping with grass in jujube orchard has a certain study. The authors should empahise on the importance and relevence of study. The results in the section of ABSTRACT should be based on digitization. And the hypothesis of this study is lacking which needs to supplement in INTRODUCTION. The paper should be by the use of clear and unambiguous professional English throughout. Check the format of the references carefully, and in the discussion it exists many mistakes. And check if the references in the section of INTRODUCTION and DISCUSSION correspond to each other in the section of REFERENCE. The figures need to revise, for instance, the coordinate headings in the diagram need to be larger.

Experimental design

(1) give the soil property base of the original jujube orchard in the paper, not in schedule form.
(2) L164, Fertilizer was applied 5-9 times during various growth stages? 5times or 9 times?
(3) L164-169, give fertilizer type for instance, how many fertilizers applied, pruning time and intensity.
(4) Kindly give the season of the experimentation along with the year.

Validity of the findings

(1) The content in the section of Results needs to be concise further.
(2) Fruit yield and quality data need to be further supplemented.
(3) The discussion should be in-depth analysis and incorporate experimental fruit yield data.
(4) Why soil nutrient content is increased in the intercropping with ryegrass compared to CK, because ryegrass is taken away as fodder for herbivorous animals and it would bring more nutrients.

·

Basic reporting

This manuscript showed that efficient planting of ryegrass in the arid sandy regions of Northern China can improve soil health as well as quality.

Experimental design

This study comes under the aim and scope of journal.
M&M: Well written but weather graph of study period may be added. Add reference for soil enzymatic activity and for formula of EN.

Validity of the findings

Although this manuscript addresses an interesting topic, there are many correction/improvement required. Comments are as below:
1. Abstract is well written but the conclusion is not clear.
2. In Introduction, background is well written but the hypothesis part should be rewritten to make it meaningful.
3. Line 85-89 “Likewise……..Weed Control”: Delete it as there is no need show this literature
4. Line 101-104: Kindly explain this. Hypothesis is not clear, kindly elaborate it.
5. In Results the presentation is very poor; Table 9 can be presented in figure.
6. Figure quality is very poor. Improve the quality.
7. In Figure 10 and 11 of correlation Analysis the significance level should be mentioned.
8. Discussion is well written but support some parts with more recent articles
9. The conclusion should be rewritten and based on the main focus of the study by keeping the
hypothesis in mind.

---

## Round 0.2 · Minor Revisions

I kindly ask you to make some minor additions to the manuscript so that it can be published:

- change the title of the article: "Effect of intercropping Lolium perenne in Ziziphus jujuba orchards on soil quality"
- figure 1: the map is distorted, the font on the map is illegible;
- figures 2-8: a space is required between the numbers and units of measurement;
- in all figures, the font size should be approximately equal to the font size in the text of the article;
- tables 3, 4: all numbers should be rounded to hundredths.

·

Basic reporting

Revised MS is now may be considered for publication.

Experimental design

M7M is well written and can be replicated anywhere.

Validity of the findings

Satisfactory after revision.

---

## Round 0.3 · Minor Revisions

Dear authors, probably (1) by mistake you attached an uncorrected version of the article or (2) you ignored most of the comments. I ask you to make changes so that your manuscript can be accepted for publication.

Specifically:

- Figures 2-8: a space is required between numbers and units of measurement;
- in all figures, the font size should be approximately equal to the font size in the text of the article

---

## Round 0.4 · accepted · Accept

Dear authors, I think that this manuscript can now be accepted for publication. You should correct the design flaws of the figures (font sizes and spaces between numbers and units of measurement in the figures) in the production phase.